

# Composition of ice particle residuals in mixed phase clouds at Jungfraujoch (Switzerland): Enrichment and depletion of particle groups relative to total aerosol

5    Stine Eriksen Hammer[1], Stephan Mertes[2], Johannes Schneider[3], Martin Ebert[1], Konrad Kandler[1], Stephan Weinbruch[1]

[1]Institute of Applied Geosciences, Darmstadt University of Technology, Schnittspahnstraße 9, 64287 Darmstadt, Germany
[2]Leibniz-Institute for Tropospheric Research, Permoserstraße 15, 04318 Leipzig, Germany
[3]Particle Chemistry Department, Max Planck Institute for Chemistry, Hahn-Meitner-Weg 1, 55128 Mainz, Germany

10   *Correspondence to:* Stine Eriksen Hammer (sehammer@geo.tu-darmstadt.de)

30



## Abstract

Ice particle residuals (IPRs) and the total aerosol particle population were sampled in parallel during mixed phase cloud events at the high altitude research station Jungfraujoch in January/February 2017. Particles were sampled on boron substrates by use of multi MINI cascade impactors operated behind an ice selective counterflow impactor (Ice-CVI) for IPRs and a heated total inlet for the total aerosol particles. Total aerosol samples were collected with a dilution setup to match the much longer sampling time behind the Ice-CVI. About 4000 particles from ten Ice-CVI samples (from seven days of cloud events at temperatures between -10 °C and -18°C) were analysed and classified with operator controlled scanning electron microscopy. Contamination particles (identified by their chemical composition) most likely originating from abrasion in the Ice-CVI and collection of secondary ice, were excluded from the further analysis. Approximately 3000 total aerosol particles from five days in clouds were also analysed. Enrichment and depletion of the different particle groups (within the IPR fraction relative to total aerosol reservoir) are presented as odds ratio relative to alumosilicate (particles only consisting of Al, Si and O), which was chosen as reference due to the large enrichment of this group relative to total aerosol and the relatively high number concentration of this group in both total aerosol and the IPR samples. Complex secondary particles and soot are the major particle groups in the total aerosol samples, but are not found in the IPR fraction and are hence strongly depleted. C-rich particles (most likely organic particles) showed a smaller enrichment compared to alumosilicates by a factor of ~20. The particle groups with similar enrichment as alumosilicate are silica, Fe-alumosilicates, Ca-rich, Ca-sulphates, sea salt and metal/ metal oxide. Other-alumosilicates - consisting of variable amounts of Na, K, Ca, Si, Al, O, Ti and Fe- are somewhat more (factor ~2) and Pb-rich more (factor ~8) enriched than alumosilicates. None of the sampled IPR groups showed a temperature or size dependence in respect to ice activity, which might be due to the limited sampling temperature interval and the similar size of the particles. Footprint plots and wind roses could explain the different total aerosol composition in one sample (carbonaceous particle emission from the urban/industrial area of Po Valley), but this did not affect the IPR composition. Taken into account the relative abundance of the particle groups in total aerosol and the ice nucleation ability, we found that silica, alumosilicates and other-alumosilicates were the most important ice particle residuals at Jungfraujoch during the mixed phase cloud events in winter 2017.





## 1    Introduction

Mixed phase clouds are important because they have an impact on the hydrological cycle, cloud electrification, and because they influence the atmospheric radiation balance (Storelvmo, 2017). Ice nucleating particles (INPs) can initiate cloud glaciation
which may cause precipitation (Myhre et al., 2013). The order of magnitude of the effect from aerosol-cloud interaction on previously called "second indirect aerosol effect" and "semi-indirect effect" is still uncertain (Myhre et al., 2013;Flato et al., 2013;Korolev et al., 2017).

In nature, spontaneous freezing of supersaturated vapour or droplets occurs at temperatures below -38°C and a relative humidity (RH) with respect to ice > ~140 % (Kanji et al., 2017), termed homogeneous ice nucleation (Vali et al., 2015). At
higher temperatures, a surface - like a particle surface - can lower the free energy and thereby assist the phase transition to ice when relative humidity allows for this, termed heterogeneous ice nucleation. Heterogeneous ice nucleation can occur in different modes; (1) deposition nucleation, (2) immersion freezing, (3) contact freezing and (4) condensation freezing. A detailed description of the different modes are found elsewhere (Vali et al., 2015;Kanji et al., 2017). Mixed phase cloud temperature range between -40°C and 0°C (Storelvmo, 2017), and the ice formation is dominated by heterogeneous nucleation
in terms of immersion and contact freezing (Lohmann and Diehl, 2006).

Ice nucleation ability has been studied off-line and on-line in many laboratory and field experiments (Hoose and Möhler (2012);Kanji et al. (2017), and references therein). Offline studies are carried out with filters or substrates for deposition nucleation, condensation or immersion freezing experiments, e.g. O'Sullivan et al. (2014);Budke and Koop (2015);Schrod et al. (2017). Online methods use different cloud chambers to quantify INPs, e.g. Möhler et al. (2003);Bundke et al.
(2008);DeMott et al. (2015);Grawe et al. (2016);Burkert-Kohn et al. (2017);Lacher et al. (2017), and some of the instruments are portable for in-situ measurements. The results from laboratory work are summarised by Hoose and Möhler (2012) showing ice nucleation efficiency of different particle groups against nucleation temperature. Summarised, biological particles seem to dominate the ice activity at higher temperature above -10 °C, whereas mineral dust is found mostly ice active below -10 °C, and organic particles and soot nucleate ice below -30 °C close to homogenous freezing. A model of ice activity worldwide in
mixed phase clouds were presented by (Hoose et al., 2010) where mineral dust represent 88% of the INPs, followed by soot (12%) and a very small fraction (<1%) of biological particles. The findings of field experiments at different locations globally are presented by Kanji et al. (2017) as function of nucleation temperature. In this paper only broadly defined classes are given to characterise the ice nucleation efficiency of INPs in different environments. To summarise, biological particles from rural areas dominate at higher temperatures (-5 °C to -20 °C), marine particles from coastal areas show a lower ice activity in the
higher temperature range than biological particles (-5 °C to -30 °C). Particles from Arctic and Antarctic locations seem to have relatively high INP abundance between -17 °C and -25 °C, and particles from areas with biomass burning show high INP concentration between -10 °C and -30 °C. Dust rich regions show particles with the highest ice activity in the range of -10 °C



to -40 °C, and these particles seem to be the most ice active component. Exact number concentration are found in Kanji et al. (2017) and references therein.

In-situ cloud measurements of ice particle residuals (IPRs) can be done with an aircraft for pure ice clouds, like for cirrus clouds, with the use of a counter flow virtual impactor (CVI) (Ogren et al., 1985). In-situ IPR sampling in mixed phase cloud

requires an extra step to separate ice crystals from droplets and is, therefore, up to now restricted to ground based measurements. A dedicated inlet system (Ice-CVI) was developed by Mertes et al. (2007) to sample freshly produced ice particles in mixed phase clouds and, after sublimating the ice, deliver the residuals (IPRs) to connected sampling or analysing instruments. Knowledge on particle groups acting as ice nuclei in mixed phase clouds is contradictory. IPRs are the residuals ice crystals formed on real INPs after they have been activated in the environment and the measured ice nucleation efficiency

of these IPRs is then considered to be the same as for INPs. This assumption may lead to some differences in reported ice nucleation ability. Another problem is the fact that different analytical techniques are used, for both IPRs and INPs. Further uncertainties arise from particle classification and the criteria used to interpret a particle group as efficient ice nuclei. For example, at the high altitude research station Jungfraujoch in Switzerland, different IPR groups were reported to act as ice nuclei. With the use of electron microscopy and looking at the enrichment relative to interstitial aerosol, Ebert et al. (2011)

interpreted complex secondary aerosol, Pb-bearing particles, and complex mixtures as ice nuclei. In contrast, Worringen et al. (2015) considered only particle groups as ice nuclei which were found with three different techniques (FINCH + PCVI, Ice-CVI and ISI). These groups included silicates, Ca-rich particles, carbonaceous particles, metal/ metal oxide and soot. Using single particle mass spectrometry, Schmidt et al. (2017) considered all particles observed in the IPR fraction as INP (biological, soil dust, minerals, sea salt/ cooking, aged material, engine exhaust, soot, lead-containing, industrial metals, NaK and others).

Kamphus et al. (2010) report mineral dust and fly ash (with and without some volatiles), metallic particles and black carbon as the most ice active particles, measured with two different mass spectrometers behind the Ice-CVI. Cozic et al. (2008a) investigated black carbon enrichment with two PSAP simultaneously behind the Ice-CVI and a total inlet, and by aerosol mass spectrometry (AMS) and single particle mass spectrometer (measuring particles between 200 nm and 2 µm) behind the Ice-CVI during cloud events. They concluded, based on the enrichment, that black carbon is ice active.

The major aims of our paper are to improve the sampling approach and to study the variation of IPRs in mixed phase clouds. In contrast to previous work (Worringen et al., 2015;Ebert et al., 2011;Kamphus et al., 2010;Schmidt et al., 2017), IPR and total aerosol were collected in parallel. This allows us to examine the ice nucleation efficiency of the various particle groups, and to investigate the dependence on temperature, particle size and air mass history.




## 2    Experimental

### 2.1. Sampling

In January/February 2017 an extensive field campaign was conducted by INUIT (Ice Nucleation Research Unit funded by the German Research Foundation DFG) at the high altitude research station Jungfraujoch in Switzerland (3580m asl). The
campaign lasted for five weeks with the aim to investigate IPRs from mixed phase clouds which are considered as the original true INPs. During mixed phase cloud events, IPRs where separated from other cloud constituents like interstitial aerosol particles, supercooled droplets and large ice aggregates by use of the Ice-CVI (Mertes et al., 2007). Total aerosol particles were sampled in parallel behind a heated inlet (Weingartner et al., 1999) to study IPR enrichment and depletion, identify contaminants and characterise the air-masses present. Total aerosol samples were collected with a dilution setup (Fig. 1) to
match the longer sampling time (up to 5 hours) of the Ice-CVI. The dilution unit is build up by two valves to control the air stream in and out of the system, making it possible to send air through two filters to dilute the incoming aerosol flow. Particles where sampled by the use of multi MINI cascade impactors with the same design as described in Ebert et al. (2016) and Schütze et al. (2017), but with the use of only one stage with a lower 50% cut-off diameter of approximately 0.1 µm (aerodynamic). The multi MINI cascade impactor is equipped with purge flow and 5 min flushing of the system was always performed prior
to sampling to avoid carryover of particles from previous samples. The IPRs were collected on boron substrates to allow detection of light elements including carbon (Choël et al., 2005;Ebert et al., 2016).

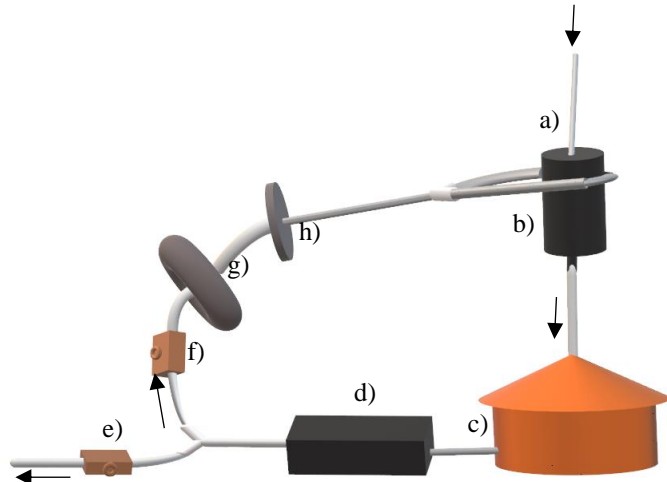

Figure 1: Illustration of the dilution unit behind the heated total inlet. a) An inlet tube attached to the total inlet, b) diluter, c) multi MINI impactor, d) pump, e) valve to control out flow, f) valve to control air going back in the system, g) pre-filter
(Whatman, Sigma-Aldrich), h) main filter (Millipore, Sigma-Aldrich). Arrows indicate the air flow direction.



### 2.2. Ice-CVI

The Ice-CVI is a modified counterflow virtual impactor which can separate freshly formed ice particles in mixed phase clouds, for details see Mertes et al. (2007). The inlet consists of several components to separate: (a) large precipitating ice crystals > 50 µm by the 90° inlet, (b) large ice particles > 20 µm with a virtual impactor, (c) supercooled droplets > 5 µm with two cold impaction plates where the droplets freeze and the ice crystals bounce off, and (d) interstitial particles < 5 µm which are removed by a counter flow virtual impactor.

### 2.3. Scanning electron microscopy

Size, morphology, chemical composition and mixing state of IPRs and total aerosol particles were investigated by scanning electron microscopy using a FEI Quanta 400 ESEM FEG instrument (FEI, Eindhoven, The Netherlands) equipped with an energy-dispersive X-ray detector (Oxford, Oxfordshire, United Kingdom). All analyses were carried out manually, referred to as operator controlled SEM, using an acceleration voltage of 15 kV and a sample chamber pressure around $1 \times 10^{-5}$ mbar. The multipoint feature "point&ID" in the Oxford software Aztec (version 3.3 SP1) was used for the operator controlled single particle analysis. On each sample, about 500 particles were measured with 5 seconds counting time for X-ray microanalysis. To ensure unbiased results all particles in an image frame with an equivalent projected area diameter ≥ 100 nm were investigated. The particles were classified based on chemical composition, mixing state, morphology and stability under the electron beam. Classification criteria are given in table 1. Particles that could not be assigned to any of the defined classes were grouped as "other". This group contains for example Mg-rich, Zn-rich and Ag containing particles. Four groups are interpreted as contamination particles: pure salt, alumina, Cu-rich and Ni-rich particles.





Table 1: Classification criteria for particle groups for both, total aerosol and ice particle residuals

| Group | Major elements | Morphology | Beam stability |
|---|---|---|---|
| Soot | C | Chain-like or more compact agglomerates of primary particles | |
| C-rich | C | No soot morphology | |
| Complex secondary particles | No X-ray spectra or S-peak | | Most particles evaporating, some relatively stable |
| Aged – sea salt | Na, S (sometimes small amount of Cl and Mg) | | Relatively stable |
| Mixed –sea salt | Na, S (sometimes small amount of Cl and Mg) + mineral composition | | |
| Ca-rich | Ca, C, O | | |
| Ca-sulphate | Ca, S,O | | |
| Silica | Si, O | | |
| Alumosilicate | Al, Si, O | | |
| Fe- alumosilicate | Al, Si, Fe, O | | |
| Other-alumosilicates | Variable amounts of Na, K, Ca, Si, Al, O, Ti and Fe | | |
| Metal/ metal oxides | Fe, O or Ti, O or Fe, Cr, Mn | Fly ash was detected as spherical particles | |
| Pb-rich | Pb, or Pb, Cl | Single particle or inclusions within particle | |
| Other | Particles which do not meet the classification criteria above | | |
| Alumina[*] | Al, O | | |
| Ni-rich[*] | Ni | | |
| Cu-rich[*] | Cu | | |
| Pure salt[*] | Na, Cl | | |

*Most likely contamination

## 2.4. Sampling days, meteorology and footprint plots

5    During seven days, ten Ice-CVI samples were taken in clouds at temperatures between -10 °C and -18 °C. Sampling day, time and temperature are presented in Fig. 2, and as table in the electronic supplement (table S1). Unfortunately, only six parallel total aerosol samples were successfully collected. The other four total samples are either overloaded or do not have enough particles on the substrate.



During the whole campaign, north-easterly and south-westerly winds were the dominating local wind directions in accordance with the topography at Jungfraujoch. Footprint plots, showing the probable air-mass residence time at the surface, were calculated with the FLEXPART model (Stohl et al., 1998;Stohl and Thomson, 1999;Stohl et al., 2005;Seibert and Frank, 2004). These plots are calculated with 10 days back trajectories and a potential emission sensitivity to determine the probable emission region of the particles arriving at Jungfraujoch. Wind roses and footprint plots are presented in the electronic supplement (Fig. S2).

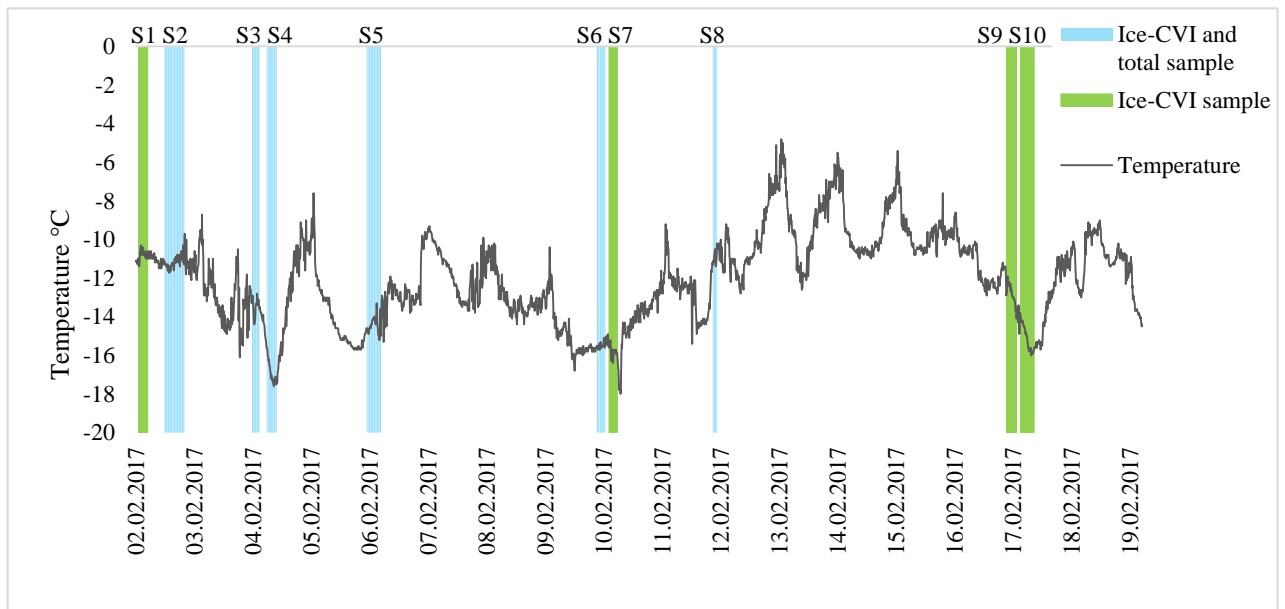

Figure 2: Temperature (°C) and sampling times in February 2017 behind Ice-CVI and total inlet. Sample numbers are given above the bars. Blue bars indicate sampling periods with parallel samples, green bars periods for which only IPR samples could be analysed. Temperature data were received from the Federal Office of Meteorology and Climatology (MeteoSwiss; www.meteoswiss.admin.ch).

### 2.3. Statistical analysis

To calculate enrichment and depletion of the different particle groups in the IPR fraction relative to total aerosol all particle group abundances are normalised to the abundance of the alumosilicate group. We have chosen this group as reference as it has the highest relative abundance in both, the IPR samples and the total aerosol. We do not show a simple ratio of proportions (e.g., proportion of alumosilicates in IPRs divided by proportion of this group in total aerosol) because the proportion is constrained to values between 0 and 1. This is generally referred to as closed data (Aitchison, 2003;Van den Boogaart and Tolosana-Delgado, 2013) and implies that only ratios of two groups can be interpreted (i.e., not the proportion of one group alone). Furthermore, we do not discuss differences in proportions between IPR and total aerosol as it was done by Ebert et al.





(2011), as this difference is strongly dependent on the relative abundance of a particle group. To overcome these problems, only alumosilicate normalised particle group abundances are used to quantify enrichment/ depletion of a particle group in the IPR fraction. This measure is termed odds ratio in the statistical literature.

The odds ratios (OR) is calculated in the following way:

$$OR_i = \frac{\left(\frac{n_i}{n_{AlSi}}\right)_{IPR}}{\left(\frac{n_i}{n_{AlSi}}\right)_{total}}, \qquad (1)$$

with $n_i$ the absolute number of particles in particle group i, $n_{AlSi}$ the absolute number of particles in the group of alumosilicates in both IPR and the total aerosol fraction. For particle groups which did not contain a single particle, one particle (which is the detection limit) was added to the respective group in order to calculate an odds ratio. For these groups the odds ratios shown in (Fig.7) represent an upper or lower limit, respectively. The odds ratios represent enrichment/depletion of a particle group normalised to alumosilicates when the IPR fraction is compared to the total aerosol. Enrichment relative to alumosilicates is discussed for each group which is present in the IPR. The two groups of complex secondary particles and soot are interpreted as depleted because these particles are not found in the IPR fraction. These two particle groups are hence depleted compared to alumosilicates and absolutely compared to total aerosol.

The Fisher test was applied to estimate confidence intervals for the odds ratio and was calculated with RStudio (RStudioTeam, 2015). Figure 4, Fig. 6 and Fig.7 are plotted in RStudio with the package "ggplot2" (Wickham, 2009). Wind roses (Fig.S2) were plotted with the RStudio package "openair" (Carlslaw and Ropkins, 2012).



## 3    Results

### 3.1. Total aerosol

Particle groups observed in the total aerosol samples include complex secondary particles, soot, C-rich, Ca-rich, Ca-sulphates, silica, alumosilicates, Fe-alumosilicates, other-alumosilicates, metal/metal oxide, sea salt (aged and mixed) and other particles
5    (Fig.3).

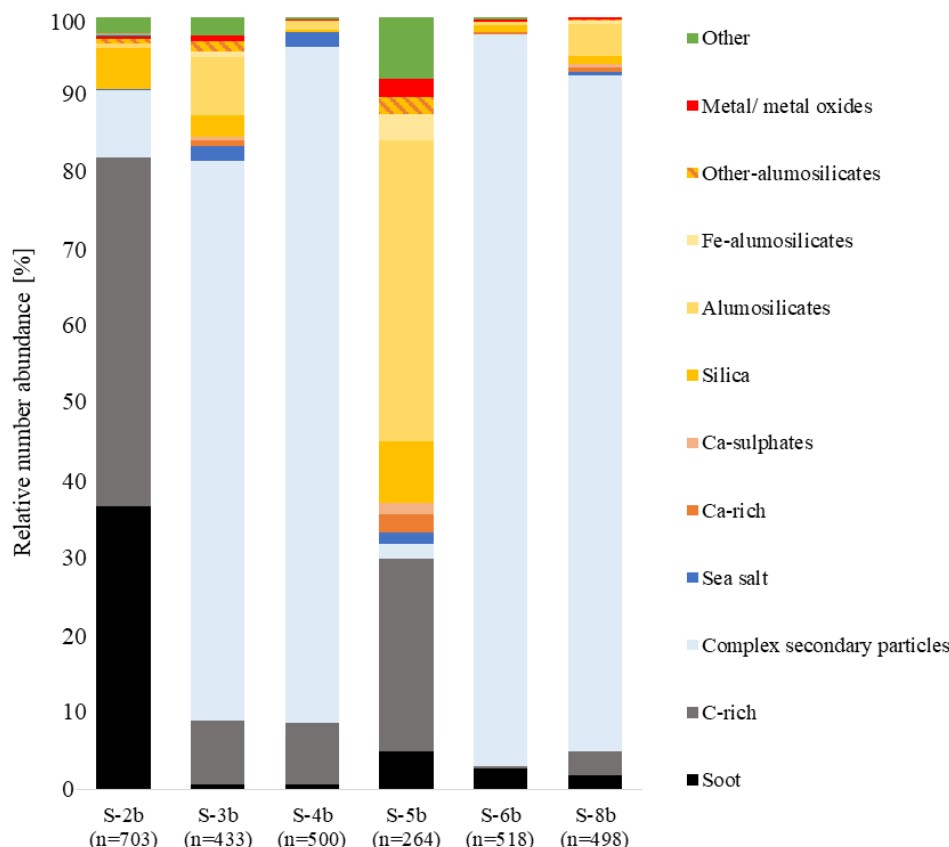

Figure 3: Relative number abundance of the different particle groups within total aerosol samples.

A few fly ash particles were detected in the metal/ metal oxides group. In addition, one group of artefact particles (Cu-rich)
10    originating from the substrate was found and excluded from further analysis. Four of the six samples are dominated by secondary aerosol, which consists of sulphates and highly instable particles (under vacuum and/or electron bombardment) for which no X-ray spectrum could be obtained. Still, remains of these particles are easily seen in the secondary electron images. The highly instable particles are classified based on the fact that they evaporated during the operator controlled X-ray analysis.



In contrast to the IPR fraction, we observed two groups of carbonaceous particles. Carbon dominated particles without typical morphology are classified as C-rich particles (Fig. S1). Chain like or more compacted agglomerates of spherical primary carbonaceous particles are interpreted as soot in accordance with previous literature e.g.,Wentzel et al. (2003);Buseck et al. (2014);Weinbruch et al. (2018). Sample S-2b is taken during night time and consists of two separate samples directly taken one after the other (for 3 hours each). The unusual high abundance of carbonaceous particles within this sample most likely result from urban/industrial sources of the Po Valley seen in the footprint plot (Fig. S2). Sample S-5b shows a high relative abundance of mineral particles which may be the result of having lost complex secondary particles in the instrument, as this sample was exposed to the vacuum of the electron microscope for a much longer time than the other samples.

Most of the total aerosol particles have geometric diameter below 500nm (Fig. 4). The mineral groups of alumosilicates, Fe-alumosilicates and other-alumosilicates are somewhat larger than the rest of the particle groups. The size distribution (dNdlogD$_p$ vs. particle diameter) is shown in the electronic supplement (Fig. S3).

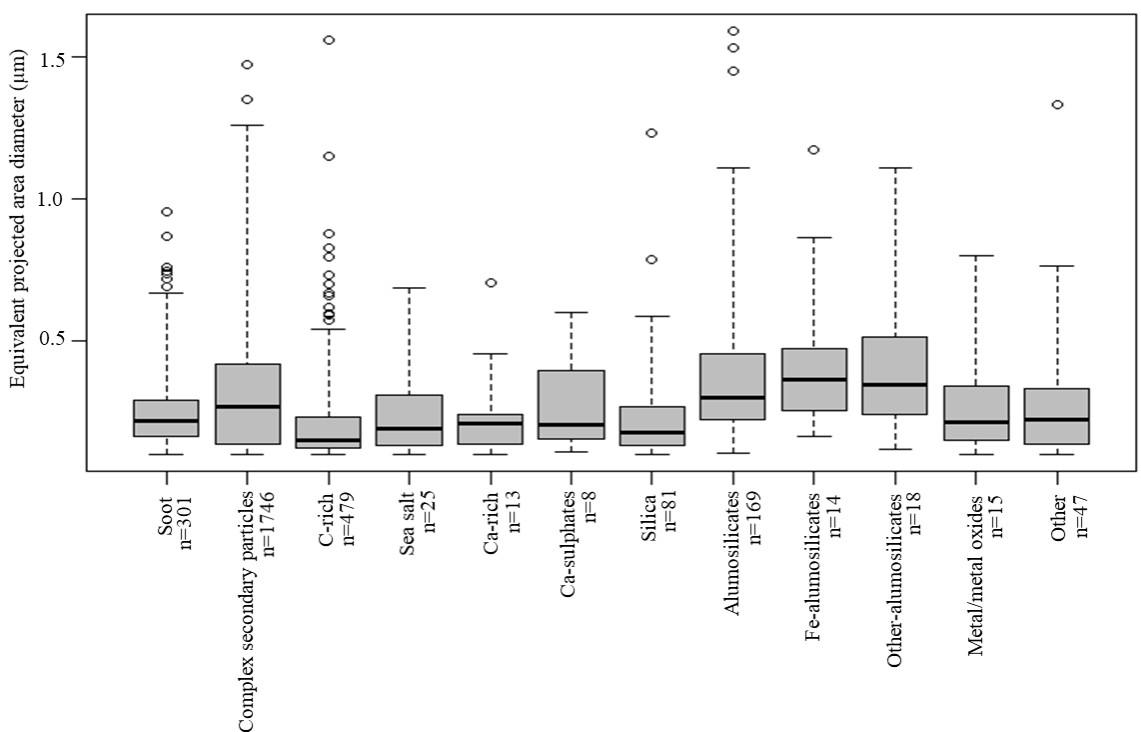

Figure 4: Size of total aerosol particles.



### 3.2. Ice particle residuals

The following particle groups were observed in the IPR samples (Fig.5): minerals (silica, alumosilicates, Fe-alumosilicates, other-alumosilicates, Ca-sulphates and Ca-rich), sea salt (aged and mixed salt), C-rich, Pb-rich, metal/ metal oxide and other particles. In addition, four groups of sampling artefacts were found: pure salt, alumina, Ni-rich and Cu-rich particles. The sampling artefacts are regarded as contamination (see discussion chapter), and are, thus not included in the figures. Composition including contamination particles is given in the electronic supplement (Fig.S5).

Mineral particles are of highest relative abundance (between 60 – 90 % by number) in all samples (Fig.5), and mainly consist of silica, alumosilicates and other-alumosilicates, as well as smaller fractions of Fe-alumosilicates, Ca-sulphates and Ca-rich particles. A small percentage ($\leq 7$ % by number) of Pb-rich particles – PbCl or particles containing heterogeneous Pb inclusions – is found in eight of the samples. Sea salt is present in all samples in variable amounts up to 12 %. The C-rich particles observed in the IPR fraction can be excluded to be soot because they do not show the typical morphology of chain-like or more compacted agglomerates of primary particles (see supplement Fig.S1). Instead, these particles are most probably organic particles. The group of metal/ metal oxide particles includes Fe-oxides/hydroxides, Ti-oxides, and steel particles (Fe, Cr, Mn alloys).

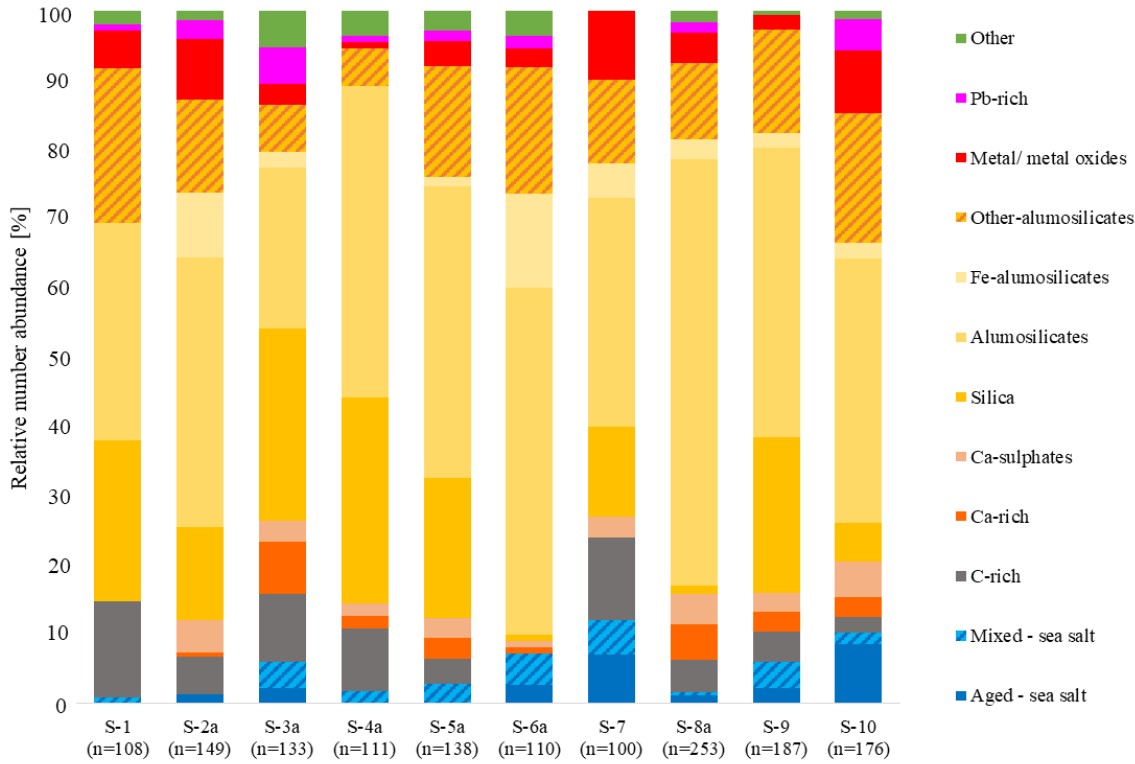

Figure 5: Relative number abundance of the different particle groups of IPR sampled in mixed phase clouds at temperatures between -10°C and -18°C. Sampling artefacts (pure salt, alumina, Ni-rich and Cu-rich particles) are not shown.




Most IPRs have an equivalent projected area diameter below 500 nm (Fig.6). The groups of Fe-alumosilicates and other-alumosilicates are somewhat larger and show a higher variation than the rest of the particle groups. The size distribution (dNdlogD$_p$ vs. particle diameter) is shown in the electronic supplement Fig.S4.

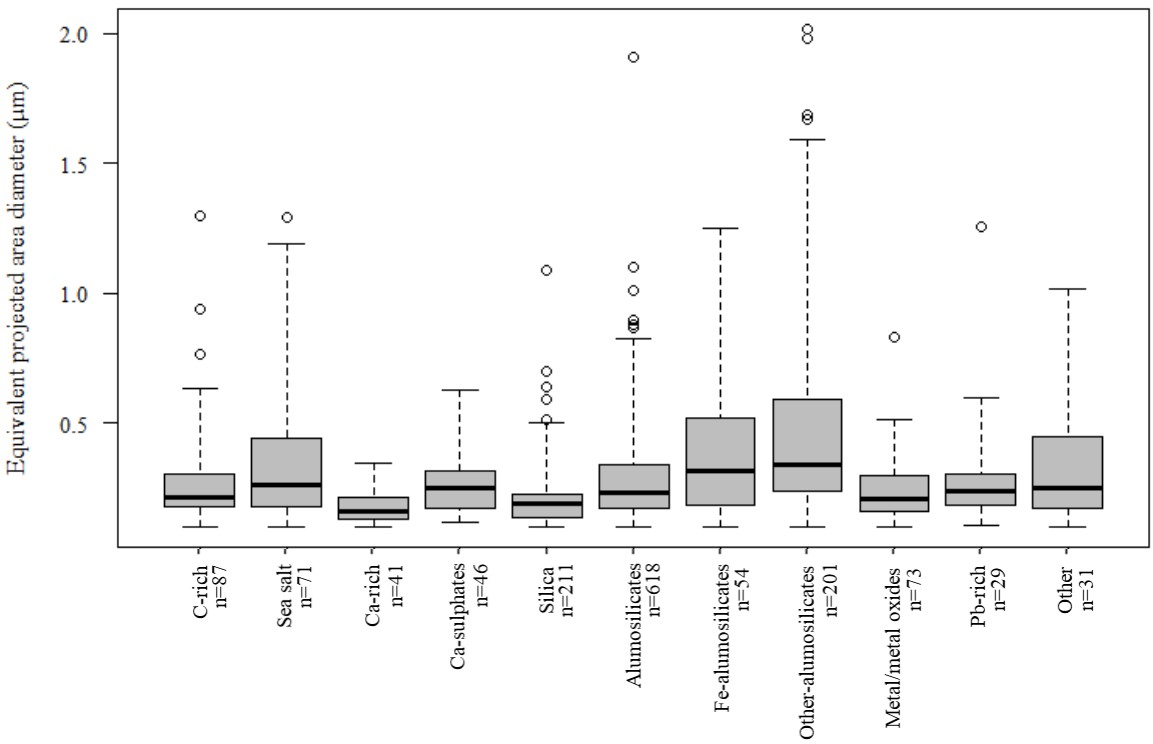

Figure 6: Size of IPRs. Three outliers of other-alumosilicates are shown presented (2.7 µm, 2.9 µm and 3.4 µm).

### 3.3. IPR versus total aerosol

For six sample pairs (simultaneous sampling of total aerosol and IPR) the enrichment/depletion of the particle groups compared to alumosilicates is shown in Fig. 7 as odds ratio. Complex secondary particles and soot are always strongly depleted in the IPR fraction, as not a single particle of both groups was observed as IPR. An upper limit for the depletion relative to alumosilicates can be obtained by setting the number of particles in the IPR fraction for both groups equal to one (the detection limit). With this assumption it can be seen that soot is depleted in the IPR fraction relative to alumosilicates by at least a factor 700, and secondary aerosol particles by a factor of at least 4200. Both particle groups are also depleted in the IPR fraction relative to total aerosol. C-rich particles are less enriched in the IPR fraction than alumosilicates by a factor of approximately 20.





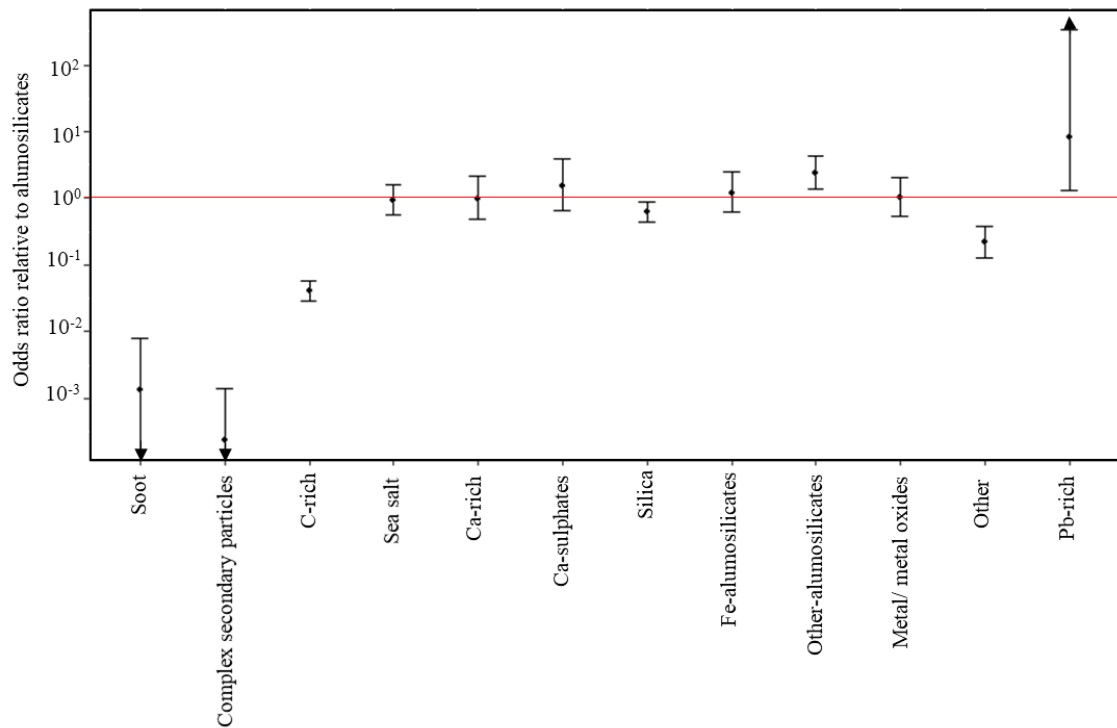

**Figure 7:** Enrichment/depletion of the different particle groups within the IPR fraction expressed as odds ratio (see text for details). The 95 % confidence interval (CI) of the odds ratio is shown as error bars. For soot and complex secondary particles the lower limit of the CI, and for Pb-rich particle the upper limit of the CI cannot be defined precisely due to counting statistics. Thus they are marked by arrows.

Pb-rich particles and other-alumosilicates are enriched (relative to alumosilicates) within the IPR fraction. However, the enrichment factor has large uncertainties due to counting statistics. The remaining particle groups are within counting error similar enriched in the IPR fraction as alumosilicates (for this latter group the odd ratio is one per definition).



## 4    Discussion

The major finding of our paper is that sea salt, Ca-rich particles, Ca-sulphates, silica, Fe-alumosilicates and metal/ metal oxides are similar ice active as alumosilicates at Jungfraujoch in warm mixed phase clouds (-10 °C to -18 °C). Other-alumosilicates and the Pb-rich particles seem to be even more ice-active as alumosilicates. In contrast, soot and complex secondary particles are strongly depleted (compared to alumosilicates and absolutely compared to total aerosol) in the ice residuals. C-rich particles are less enriched than alumosilicates by a factor of approximately 20. Thus, it is concluded that their ice nucleation ability under these conditions is significantly lower. The ice nucleation activities of the different particle groups are discussed intensely in chapter 4.3.

### 4.1. Methodological problems

#### 4.1.1.    Sampling artefacts

The observed alumina, pure salt, Ni-rich and Cu-rich particles are regarded as sampling artefacts. Alumina particles are found in all IPR samples at relative high number abundances between 25 and 70 %, despite the fact that the Ice-CVI was coated before the present campaign with Ni to avoid this contamination. The relative abundance of alumina particles in IPR samples is higher in our campaign compared to two previous campaigns at Jungfraujoch using the same instrumentation nut without the Ni coating of the Ice-CVI (Ebert et al., 2011;Worringen et al., 2015), but this might be explained by the fact that we only focused on fine particles. In contrast to previous work, we sampled IPR and total aerosol in parallel to be able to clearly distinguish instrumental artefacts from IPRs. As we did not detect a single alumina particle in total aerosol samples, this particle group is regarded as contamination. Alumina particles are easily recognised and were subtracted from the real IPRs. Nevertheless, their presence helped substantially to locate the impaction spot on the boron substrates.

Secondary ice processes can produce ice crystals in the critical size range selected by the Ice-CVI. The low temperature during sampling does not support the Hallett-Mossop process (Hallett and Mossop, 1974) regarding rime splintering, but other secondary processes producing ice crystals like ice-crystal break up, blown snow or crystal-crystal collisions in the critical size range are plausible (Mertes et al., 2007). We hypothesise that pure salt is an artefact due to sampling of the mentioned secondary ice production processes in clouds. The presence of sodium and chloride in ice crystals previously acting as cloud condensation nuclei can later form solid NaCl in line or on the substrate after evaporation of water. This hypothesis is inconclusive because pure salt is not observed in the total aerosol fraction, where only aged and mixed salt are present. This might be explained by evaporation of ice crystals in the heated inlet and the longer sampling line, and the relatively low number concentration of these particles compared to the dominating groups (soot and complex secondary particles) in the total aerosol samples. It should be mentioned here that sea salt was considered to be an artefact in the IPR fraction by Worringen et al. (2015).



A few Ni-rich particles (1-7% relative by numbers) were encountered in the IPR fraction but not in the total aerosol. The Ni-rich particles most likely stem from the Ni coating of some parts of the Ice-CVI. The few Cu-rich particles found, in both total aerosol and the IPR samples, are from the boron substrate where boron is embedded in cupper.

### 4.1.2. Accuracy of particle group abundance

Accuracy of the particle group abundance depends on three different factors: (1) separation of IPR from the rest of the aerosol particles by the Ice-CVI and deposition losses behind both inlets, (2) detection of particles in SEM, and (3) the classification procedure. Sampling issues like abrasion, deposition losses and ice crystal breakup may occur in the Ice-CVI (Mertes et al., 2007). Abrasion particles were easily recognised as discussed in the previous paragraph. Sampling of secondary ice may have led to the relative high abundance (~2 – 28 %) of pure salt particles in the IPR fraction discussed in the previous paragraph. As we regard pure salt particles as artefact, they are not included in the sea salt group. Deposition loss can generally not be excluded. Three of the total aerosol samples (S-3b, S-4b and S-6b) are sampled under conditions where the concentration of the total inlet was lower than the interstitial inlet. There are two possible explanations for this: deposition loss in the total aerosol inlet and/or a leak in the interstitial inlet. The relative abundance of the different particle groups in these samples is however comparable to previous findings at Jungfraujoch (Cozic et al., 2008b;Kamphus et al., 2010;Fröhlich et al., 2015). A possible deposition loss leading to systematic bias in the concentration measurements does not seem to change the relative abundance of the different particle groups. Our conclusions are thus not affected as we do not discuss number concentrations.

For most particle groups we do not expect to have significant detection artefacts in SEM. These particle groups are detected with high efficiency, in both the total aerosol as well as the IPR fraction. However, C-rich particles and soot may be interchanged in total aerosol samples because the image quality can be reduced by evaporating complex secondary particles leading to less efficient detection of carbonaceous species which have a low contrast in SEM images. Usually, evaporation of complex secondary particles is not a problem because the particles are observed at the start of analysis. Nevertheless, in one sample, complex secondary particles were lost prior to observation because this sample was erroneously left in the chamber for a longer time before it was analysed. However, these effects seem to be small because we have observed a comparable abundance of carbonaceous particles and complex secondary aerosol particles (in total aerosol) as in previous work (Cozic et al., 2008b).

The classification criteria used (table 1) may lead to problems for small (below approximately 150 nm equivalent projected area diameter) carbonaceous particles. Due to the limited lateral resolution of the instrument, the typical morphology of soot may not be recognised for small particles. In this case, soot would be misclassified as C-rich particles. Still, the sum of both particle groups should be accurate. However, this problem is only significant for the total aerosol samples because evaporating secondary aerosol in these samples leads to deterioration of the image quality. Misclassification of soot as C-rich particle would imply that soot is even more depleted in the IPR fraction.





### 4.2. Composition of total aerosol

Four of the six total aerosol samples are dominated by complex secondary particles (Fig.3), which seems to be typical for Jungfraujoch (Cozic et al., 2008b;Fröhlich et al., 2015). Two samples (S-2b and S-5b) have a different composition (Fig.3).

The first sample (S-2b) shows a higher carbonaceous fraction, the second sample (S-5b) a higher fraction of mineral particles and C-rich particles. The high soot and C-rich particle abundance of the first sample may be explained by footprint plots showing that the air-mass had a longer surface residence time over Po Valley (Italy) which is an urban/industrial area with abundant sources of carbonaceous particles. The second sample with unusual composition may suffer from an analytical artefact. Complex secondary particles in this sample may be lost in the electron microscope due to a much longer analysis time

leading – of course – to a larger relative abundance of the remaining particle groups. However, this potential artefact does not influence the enrichment factors of all other particle groups. Merely, the odds ratio of complex secondary particles shown in Fig.7 will be somewhat lower. Our general conclusion that complex secondary particles are inefficient ice nuclei under the investigated conditions is not changed.

Most particles of the total aerosol have sizes below approximately 1µm which is in good agreement with Herrmann et al.
15  (2015).

Overall, our total aerosol samples consist of complex secondary particles (60 % by number) and C-rich particles (16 %), soot (10 %) and mineral particles (14 %). This composition is similar to previous findings at Jungfraujoch during winter. According to Cozic et al. (2008b) the total aerosol is dominated by organic matter and secondary aerosol (87 % by mass), and smaller contributions of black carbon (4 %) and a "non-determined mass" (reported as *"assumed to be composed of insoluble*
*compounds such as silicate from mineral dust"*) fraction (9%). It was also shown by Kamphus et al. (2010) that the main components of the ambient aerosol at Jungfraujoch in winter (2007) are sulphate and organics, and only a small fraction (between 1% and 17%) is classified as mineral particles.

With respect to ice nucleation, mineral dust particles are of most importance (see chapter 4.3.). Alumosilicates are the most abundant group of mineral particles in the total aerosol with almost twice the amount of silica. This fits well to the distribution
of different minerals in soils presented by Hoose et al. (2008) where kaolinite and illite show a higher abundance than calcite and quartz in the clay fraction worldwide. Other-alumosilicates and Ca-rich particles are present in four of the six samples at low number concentration (1-2 %). Ca-containing particles at Jungfraujoch were also found by Cozic et al. (2008b), albeit mainly in the coarse mode.

The footprint plots (Fig.S2) were quite similar with high particle residence time over the North Atlantic Ocean. None of the
samples are taken during mineral dust events, which normally occur in spring at Jungfraujoch (Coen et al., 2007). One total aerosol sample with higher fraction of carbonaceous particles had a higher surface residence time over Po Valley than the rest.





### 4.3. Ice nucleation activity of different particle groups

Ice particle residuals mainly consist of mineral particles (Fig.5). The classes of Fe-alumosilicates, Ca-sulphates, Ca-rich, silica, sea salt and metal/ metal oxides are similar enriched as alumosilicates (odds ratio ~1). Other-alumosilicates are more enriched than alumosilicates by a factor of ~2. The mineral particles abundance between 60-85% in the IPR fraction is in good agreement with previous findings for mixed phase clouds at Jungfraujoch (Kamphus et al., 2010;Ebert et al., 2011;Worringen et al., 2015). Mineral particles are also reported as ice active in cirrus clouds (DeMott et al., 2003;Cziczo and Froyd, 2014). Studies of IPRs in cirrus clouds are mentioned sometimes in the discussion to show which kind of INPs are found in the environment, independent on the cloud regime. It has to be emphasised here that this is not meant as direct comparison as the temperature and freezing regimes are quite different, note that deposition nucleation dominates in cirrus clouds (Cziczo et al., 2013).

The size of IPRs varies between the detection limit (100 nm) and 3.4 µm (Fig.6). The size distribution is comparable to previous findings by Worringen et al. (2015) showing a maximum around 300 nm. We did not find a relationship between the size of the particles and the enrichment factor (odds ratio) presumably because the particles size did not differ much.

The sampling temperature varied between -10°C and -18°C (Fig.2). Temperature was measured at the station, and can differ to the onset ice nucleation temperature of the particles depending on where in the mixed phase cloud nucleation occurred. None of the particle group abundances in the IPR fraction showed a systematic temperature dependence. However, based on the limited number of samples and the relatively small temperature range, no definite conclusion regarding the temperature dependence can be drawn.

The importance of a given particle group for ice nucleation in the atmosphere depends on the ice nucleation ability and the abundance of this group in the total aerosol. Both parameters will be discussed in the following. Complex secondary aerosol particles and soot were not found in the IPR fraction, in contrast to previous work at Jungfraujoch (Cozic et al., 2008a;Ebert et al., 2011;Worringen et al., 2015;Schmidt et al., 2017), even though these groups dominate the total aerosol fraction. Thus, their ice nucleating ability can be assumed to be very low. In the present study, complex secondary particles are defined by the presence of a S-peak in the X-ray spectrum and/or the instability under electron bombardment. It must be emphasised here that this particle group most likely also consists of a substantial fraction of organics and nitrates (Vester et al., 2007).

C-rich particles were observed in the total aerosol and the IPR fraction, but are less ice active than alumosilicates (odds ratio ~ 0.04). The C-rich particles encountered are most likely organic aerosol. This is in good accordance with the results of many field studies (see the recent review by Knopf et al. (2018) and references therein) which show that organic aerosol is found in the IPR fraction, but is depleted relative to total aerosol.

Alumosilicates are enriched in all samples and have the highest relative number abundance in the IPR fraction. Alumosilicates are also found to be efficient ice nuclei in other field experiments (Cziczo et al., 2013;Worringen et al., 2015;Iwata and Matsuki,



2018). Among alumosilicates, kaolinite is reported as efficient ice nucleus in laboratory studies (Zimmermann et al., 2007;Murray et al., 2011;Wex et al., 2014;Freedman, 2015). As alumosilicates often have a high abundance in the total aerosol and in the IPR samples, they are the most important particle group for ice nucleation. Therefore the enrichment/depletion of the particle groups was normalised to this group.

5    Silica is the second most abundant mineral particle group in the IPR samples and the only mineral group which seems to have a somewhat lower ice activity than alumosilicates (upper limit of 95 % confidence interval of the odds ratio <1). However, keeping in mind the counting error for alumosilicates, silica is statistically similar enriched. This observation is in agreement with Atkinson et al. (2013) but in contradiction to Eastwood et al. (2008) who concluded that quartz is less ice active than kaolinite and montmorillonite. The silica fraction in the IPR samples varies between 1 and 30 %. Boose et al. (2016) point out 10    that quartz is always present in atmospheric dust in all size ranges, even in the smallest size fraction which is dominated by clay minerals. They conclude that quartz is an important atmospheric INP component because it is present in the size fraction with the longest atmospheric residence time. Despite the fact that the enrichment of silica is somewhat lower than alumosilicate, the relative high abundance in the IPR fraction in our samples confirms this conclusion.

Fe-alumosilicates are similar enriched in the IPR fraction as alumosilicates. Fe-alumosilicates were reported as cloud residual 15    by Matsuki et al. (2010). As these authors did not differentiate between droplet and ice crystals, nothing can be said about the ice nucleation ability of Fe-alumosilicates. This mineral group is not present at high relative abundance at Jungfraujoch, thus, it will not contribute much to ice nucleation at this location.

The group of other-alumosilicates most likely consists of different minerals like for example feldspars, illite and smectite. Laboratory studies (Atkinson et al., 2013;Iwata and Matsuki, 2018) showed that K-feldspar and clay minerals (Zimmermann 20    et al., 2008;Hiranuma et al., 2015;Boose et al., 2016) have a high ice nucleation ability compared to other minerals. A high ice nucleation ability of clay minerals is also reported from field experiments (Targino et al., 2006;Worringen et al., 2015). Also our field study shows an enrichment of other-alumosilicates in the IPR fraction indicating a high ice nucleation ability. However, as feldspar is less common in the smallest dust fraction, it was concluded by Boose et al. (2016) that at least the feldspar group is generally of minor importance.

25    Ca-rich and Ca-sulphate particles are relatively low in number concentration, both in total aerosol and IPR samples. Similar to quartz, calcium containing particles showed different ice nucleation ability in previously laboratory studies (Zimmermann et al., 2008;Atkinson et al., 2013). In field experiments, however, Ca-rich particles and Ca-sulphates were observed in the IPR fraction (Ebert et al., 2011;Worringen et al., 2015;Iwata and Matsuki, 2018).

Based on chemistry, three subgroups of salt can be distinguished in the IPR samples: pure salt, aged-sea salt and mixed-sea 30    salt. The pure salt is regarded as artefact (see chapter 4.1.1.) and, thus, excluded from the further analysis. Due to their low number abundance, the two other salt subgroups are combined into the sea salt group. Sea salt is similar enriched as



alumosilicates. The ice activity of salt and sea salt is still controversial due to discrepancies between different laboratory studies (Wise et al., 2012;Niehaus and Cantrell, 2015;Ladino et al., 2016). Kanji et al. (2017) assign these differences to the experimental setup, i.e. different size, composition and particle generation methods. In field experiments, however, salts are present in the IPR fraction of both, cirrus and mixed phase clouds (Targino et al., 2006;Ebert et al., 2011;Cziczo et al.,

2013;Worringen et al., 2015;Iwata and Matsuki, 2018). It is advocated by Iwata and Matsuki (2018) that pure NaCl is not ice active due to molar depression of the freezing point. However, sea salt may act as an INP due to the presence of organic particles in sea spray (Wilson et al., 2015;DeMott et al., 2016;Iwata and Matsuki, 2018). We cannot prove with our measurements if the ice activity is due to organic species within the salt fraction, and hence also not exclude this hypothesis.

The enrichment of metal and metal oxides is similar to alumosilicates. The ice activity of different metal and metal oxide

particles varies with their chemical composition (Kanji et al., 2017). Our samples are dominated by FeCrMn (steel), Ti-oxide, Fe-oxide. Literature regarding the metal/ metal oxide group is ambiguous. Hematite was reported as ice active by Zimmermann et al. (2008). In contrast, hematite, magnetite and rutile were found not to be very ice active in deposition mode by Yakobi-Hancock et al. (2013). Even so, metal and metal oxides are often found in IPR samples from cirrus and mixed phase clouds (Kamphus et al., 2010;DeMott et al., 2003;Ebert et al., 2011;Worringen et al., 2015;Schmidt et al., 2017).

Pb-containing particles are present in the IPR fraction as already reported in previous work at Jungfraujoch (Cziczo et al., 2009;Kamphus et al., 2010;Ebert et al., 2011;Worringen et al., 2015;Schmidt et al., 2017). In the present study, Pb-rich particles are the most enriched particle group. A high enrichment of Pb-rich particles among IPRs was also reported by Ebert et al. (2011). In addition, laboratory work showed that Pb can increase the ice activity of mineral particles considerably (Cziczo et al., 2009;Yakobi-Hancock et al., 2013). Helicopters and small aircrafts were discussed as local sources of Pb at Jungfraujoch

by Kamphus et al. (2010) and Ebert et al. (2011). As the samples were collected during in-cloud conditions, we do not expect freshly on-site emitted Pb-rich particles from the mentioned sources. A time delay between emission and sampling results in relatively low concentrations of Pb in the ambient air in clouds at Jungfraujoch. However, Kamphus et al. (2010) and Schmidt et al. (2017) detected Pb-bearing particles with mass spectrometry in both ambient air and IPRs. Keeping in mind the better counting statistics of mass spectrometry, it seems plausible that the total aerosol contains a small amount of Pb-rich particles

which were missed in our total samples.

To summarise, the two particle groups of complex secondary particles and soot are strongly depleted compared to alumosilicates as well as absolutely to the total aerosol. Despite an uncertainty due to potential misclassification, the C-rich group is less enriched compared to alumosilicates. Other-alumosilicates and Pb-rich particles are enriched compared to alumosilicates. A high enrichment of Pb-rich particles indicates that this group is more ice active than the rest of groups present

in the IPR fraction. All other particle groups (silica, Fe-alumosilicates, Ca-sulphates, Ca-rich, sea salt and metal/metal oxides) are similar enriched as alumosilicate. The relative high abundance of artefacts was identified by comparing the IPR and total aerosol fraction, showing how important parallel sampling is for identification of IPRs. Taken into account the relative



abundance of the particle groups in total aerosol and the ice nucleation ability, we conclude that silica, alumosilicates and other-alumosilicates were the most important ice nucleating particles in mixed phase clouds between -10 °C and -18 °C during the campaign at Jungfraujoch in winter 2017.





*Data availability:* The data set is available for the community and can be accessed by request to Stine Eriksen Hammer (sehammer@geo.tu-darmstadt.de) of the Technical University Darmstadt.

*Competing interests:* The authors declare that they have no conflict of interest.

**Acknowledgments**

5   Stine Eriksen Hammer would like to thank Annette Worringen and Nathalie Benker for discussion and support, and Thomas Dirsch for building the dilution unit. We thank the whole INUIT-JFJ team for discussions and support. The authors thank MeteoSwiss for meteorological data, and the International Foundation HFSJG who made it possible to carry out the experiment at the high altitude research station Jungfraujoch. The authors also gratefully acknowledge the German Research Foundation for financial support within the research group INUIT - INUIT (FOR 1525) and within grant KA 2280/2-1.

This project has received funding from the European Union's Horizon 2020 research and innovation programme under grant agreement No 654109.

Author contributions

Stine Eriksen Hammer collected the samples, analysed the particles by electron microscopy, performed data analysis and prepared the manuscript. Martin Ebert contributed to electron microscopy and data analysis. Konrad Kandler designed the

15   dilution unit and contributed to data analysis. Stephan Mertes designed, improved and operated the Ice-CVI during the campaign. Johannes Schneider organized the field campaign at Jungfraujoch and contributed to data analysis. Stephan Weinbruch contributed to data analysis and manuscript preparation.



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
