# Peer review of "Composition of ice particle residuals in mixed phase clouds at Jungfraujoch (Switzerland): Enrichment and depletion of particle groups relative to total aerosol"

_Atmospheric Chemistry and Physics, 2018_

## Referee Comment (RC1) · Anonymous Referee #1 · 17 Jun 2018

Hammer et al. consider the properties of particles within ice crystals, aka ice particle residuals (IPRs), at the Jungfraujoch, a high altitude mountain site in Europe. For this work they use a published ice selective inlet aka 'Ice-CVI', and compare this to the total aerosol via a simultaneous heated inlet sample to provide all, aka 'total', particles. In total, they consider ~4k ICRs over 7 days and in the ~-10 to -20 deg C range (i.e., mixed phase clouds). These particles were deposited on substrates for off-line analysis with electron microscopy (EM).

There are several papers over the last decade, many by the groups represented here,

describing such measurements at the Jungfraujoch so this is not altogether novel. It does represent important data in an area of atmospheric science that is currently in need of more information. My major concern, however, is that the way ice residuals are described seems very simplistic and not of use to the broader atmospheric science community. For example, what is a C-rich particle? How would someone using AMS or a modeler compare this to their understanding of the atmosphere? Since studies of this type have been published, again, many by this group, I believe they here have a responsibility to make the results more comparable, and therefore more useful, to the broader community.

I therefore suggest the authors consider the following suggestions and, pending another review, that this paper could be published in ACP.

1. The literature seems to predominantly use the term 'ice residual' (IR) as opposed to IPR. Is there a reason the authors have suggested a new term? Is it different than IR? If so a comprehensive description – and difference from IR - needs to be made. As it reads it seems they are the same so, if there is no difference, could you please remain consistent with the literature term IR so as not to confuse the reader. 2. Page 3 Line 8 : The authors seem to suggest water vapour homogeneously nucleates ice at -38 deg C and 140% RH; but that is not correct. One must first have droplets. Please remove 'vapour or' from this sentence. 3. Page 5 Line 5&6 : From the intro, the authors 'assume' the IPRs are INPs, not 'consider'. This is a very important distinction. The authors discuss artifacts; thus they themselves show they can not make the direct association implied by 'consider'. 4. Table 1 : Most aerosol composition measurements show the presence of sulfate and SOA particles as a dominant, if not the dominant, aerosol, at least by number. Is the 'complex secondary particles' this class? This seems to be implied later in the Discussion but is never clearly stated. If so, even if it is an assumption, this needs to be stated for clarity and comparability to the literature. (note: this seems to be suggested on Page 10 but warrants more than 1 line of text). 5. Similarly, biomass burning particles are often noted as being of abundance just

below sulfates and SOA. Is this the C-rich class here? If so please state this. I stress here: while the authors are reporting what they observe with their EM technique, they also need to make it clear their classes relate to common particles types if they wish to publish in a journal such as ACP. Please make these comparisons. 6. Similarly, what are Ca-rich particles? The discussion seems to suggest they are mineral dust? How are these associated with any common aerosol type? 7. The authors discuss observations of biological IRs in the introduction. These don't appear on Table 1 or Figures 4 or 5. Are they not observed or is there an instrumental reason they can't be detected? Are they a subset of the C-rich category? This needs to be stated clearly in the paper, especially in Figure 5, as it goes directly to comparison to the literature on ice residual composition and in the location where Table 1 is described. 8. Page 7, Line 6. Please remove 'Unfortunately, only'. Data are what was collected. 9. Sample S-2b looks rather like a local combustion event. This seems to be implied later in the discussion. If it is please state at the location of the Figure for clarity to the reader. 10. Given this, could you please present figure S2 at this location – move it into the main paper from the supplement - for the samples to give the origin some context? It is mentioned in the text in the discussion but really needs to be given at this location. 11. Sample S-5b looks rather like a mixed mineral and perhaps combustion event. Is this so? It again does not look like a clean troposphere. Is this so? If it is please state at the location of the Figure for clarity to the reader. Please also see last comment re: back-trajectories. 12. I am confused with Section 4.1, 'Methodological Problems'. I believe this all needs to do in the Experimental section (2), not held until after the data is presented. First, Figure S5 seems to indicate that most of the collected particles are artefact. Is this the case? If so please state the percentage in the Artefact section. Second the reader goes through the results but only after they are presented learns there are issues with the inlet and perhaps the EM data which appear to outnumber the real data by several factors. This is not a logical order. This all needs to be clearly stated and placed in Section 2, not held to Section 4.

---

## Referee Comment (RC2) · Anonymous Referee #2 · 9 Jul 2018

**General Comments**

This paper discusses the most recent evaluation of the composition of ice crystal residues as measured by an ice-selective counterflow virtual impactor inlet (CVI) at the Jungfraujoch experiment station. The broader community is surely interested in this work, in general and to understand if all measurement artifacts have been solved to the point that a consistent and informative data set can be collected regarding the source of ice nucleating particles and other information on microphysical processes in winter clouds at this site. The short answer based on this paper is that, while some

issues have been resolved, this remains a work in progress. While a potentially useful paper, this one could have used stronger editing prior to submission. The overall organization is generally good, but the discussion bounces around such that the various facts are not subjected to a structured discussion. As for specific critical revisions needed, a vital one is to bring forward the fact that a major measurement artefact remains unresolved, that of Al in many particles. I saw no clear discussion of the potential source for this contamination. Otherwise, I appreciated the attempt to categorize electron microscopy microprobe data, although I stumbled on the categorization of "sea salt", by which was meant an assortment of possibilities. This pointed to a general need to be more descriptive about the categorizations and how artefacts were defined in comparison to specific sources. With revision, this paper will become acceptable for publication, although it remains another step in the direction of attempts to extract information on ice nucleation processes through inspection of ice particles residuals.

Specific questions/comments for potentially addressing are listed below.

**Specific Comments**

*Abstract*

A few details here should be clarified. 1) I have no idea what a multi MINI impactor is, but it certainly does not need listing in the abstract. Just the basic technique should be stated. 2) It is not clear how a dilution setup allows for matching a total aerosol sample to the Ice-CVI sample. Can this be explained in plain language? Perhaps, "A dilution system was used to collect total particles at a similar rate to Ice-CVI particle collections", although I do not know how that is managed, and it seems that it failed about 503) State temperatures as "local" or "site". These are not necessarily the cloud activation temperatures. 4) "Approximately 3000 total aerosol particles from five days in clouds were also analysed." Is this referring to IPRs or to interstitial particles, or to all non-ice particles?

*Introduction*

Page 3, line 8 – Spontaneous freezing of supersaturated vapour? In the Earth's atmosphere? I have not seen such a statement in the literature in some time. Remove unless you can support the feasibility of any process other than homogeneous freezing, not "homogeneous ice nucleation".

Page 3, lines 12 to 15 – Add "hypothesized" to modes. The last sentence is repetitive with regard to mixed-phase temperatures where heterogeneous nucleation is the source of ice initiation.

Page 3, paragraph 2 – The need for this thesis-type material is questionable. I suggest to revise and reduce or even omit most of this and get straight to the point, which seems to be that information on the relevance and importance of different ice nucleating particle types has come from laboratory measurement, and these emphasize the importance of mineral dust particles except at very modest cloud supercooling. What this paragraph does not seem to mention are specific studies where activated ice nucleating particles have been studied for composition, not simply tested as single collected types in the laboratory.

Page 4, line 4: Ogren et al. (1985) is not in the reference list. There is a substantial amount of literature since in which airborne CVIs have focused on ice clouds, and Cziczo, Froyd and colleagues have emphasized some other constraints on ice cloud sampling of IPRs (e.g., a focus on small ice, as done also in this study – line 6 statement, although no indication is given as to why fresh ice is needed). While an aside of sorts, the utility of sampling in mixed-phase clouds for ice nucleation studies using a non-ice-CVI is not reflected here, since the focus is on IPRs. The fact that one gets both IPRs and liquid cloud residuals when sampling in mixed-phase clouds is not necessarily a detriment, and this makes it suitable for ice nucleation measurements and subsequent collection of the activated INPs for compositional analyses (already mentioned in the preceding point). This is alluded to later in mentioning use of a FINCH for a similar purpose.

Page 4, lines 8-10: But then this introduction is followed with these lines, which I could not understand - "Knowledge on particle groups acting as ice nuclei in mixed phase clouds is contradictory. IPRs are the residuals ice crystals formed on real INPs after they have been activated in the environment and the measured ice nucleation efficiency of these IPRs is then considered to be the same as for INPs." I expected the first sentence to be immediately supported. Is this a new paragraph? It is not a good one for sure. Rewrite it to be concise, and get to that point. Is the contradiction mentioned referring only to studies done at Jungfraujoch, or what other studies? Will this study seek to resolve contradictions? What is a "real" INP? I suggest to remove this terminology. I think I understand the last part to mean that the composition of IPRs is considered to be those of INPs that were active at the local temperature of observation.

*Experimental*

Page 5, lines 5-6: Please explain or omit the statement "…original true INPs." You will simply assume that IPRs represent INPs active at the cloud temperature of observation, correct? Are you trying to infer that other methods will not detect INPs? I think you are trying to say that the residuals reflect INPs that were activated in the cloud. But are you saying that every ice crystal contains an INP? I do not think that can be supported, if for example secondary ice formation processes were active.

Page 5, line 7: typo, "were" not "where"

Page 5, lines 9-10: Can you explain the need for dilution of the total aerosol sample a little better? i.e., there would be too many particles if collected for the entire time period?

Page 6, lines 18-19: Why are pure salt, alumina, Cu-rich and Ni-rich particles considered as contamination? It would be nice to consolidate this information in one place. In the end, no source is identified or even suggested for the alumina particles assumed as contamination, and I find the fresh salt explanation to be questionable. I gather later that the Al is assumed to come from ice crystals striking the walls of the CVI, despite

coating them with Ni, but it is almost incomprehensible how this contamination exceeds that found in any previous study (page 15).

Page 7, line 5: This seems to require a statement that the cloud sampling temperatures were considered as appropriate as the ice crystal formation temperature. Could satellite data say anything about coldest cloud top temperatures at these times? Or do you also assume that the limited ice crystal size range sampled restricts this condition?

*Results*

Page 12: A general comment - it might be nice to show both a representative particle image and elemental spectra for each of the different particle composition categories. This could go in the supplement in addition to the single example given.

Page 12: General comment 2 – It is only if one goes immediately to look at Fig. S5 at this point that one realizes that the vast majority of particles were categorized as artifacts. Surely this needs to be mentioned upfront. Greater that 50

Page 12, line 7: I wonder if in the basic analysis performed if a mineral particle could be distinguished as being from desert or from other soils? I assume this would remain unresolved, since the soil particle could have multiple potential actual ice nucleation sources, including trace organics.

Page 12, line 10: When the authors say "sea salt", what is meant? Is it only NaCl, or does this refer to aerosols of sea spray origin, with a more complex mixing state? There are only two categories, aged and (aged-) mixed, and by mixed are also included mixtures with other aerosols such as minerals. This makes attribution specifically to "sea salt" nebulous, and yet statements are subsequently made in the results about the ice activity of "sea salt". This is problematic.

Page 12, line 14: I assume that aluminum oxides are omitted from the metal oxide category because of the alumina contamination that is not really discussed?

*Discussion*

Page 15, line 1: Sea salt is similarly ice active as aluminosilicates? Is it the sea salt, the organic content of marine aerosols, or the particles they are mixed with? Hence my earlier question. Perhaps these should be stated to be sea salt-containing particles, and a statement is needed about how this does not identify the "salt" as the ice nucleating component.

Page 15, lines 15-17: A focus on fine particles is mentioned as an explanation for the occurrence of more alumina in this study, apparently from crystals etching this from the CVI walls (nowhere stated clearly). This is the first mention of any different focus in this study. What is meant be a focus on fine particles? Why would there be so much less Ni and so much more Al? Was the coating quickly destroyed? Ineffective? Also on line 15, "but" is misspelled.

Page 15, lines 24-26: If pure or "fresh" salt is an artefactual reflection of secondary ice formation contributions, how is this reliably distinguished from sea spray aerosols? Would aging of sea salt always occur for marine particles reaching the site? Relatively unaged marine aerosols are found at other remote locations.

Page 16, line 12: By "concentration of the total inlet" do you mean the accumulated particle number concentrations sampled from the total particle inlet (after dilution)?

Page 17, lines 8-10: This might well be the third mention of the sample that was exposed to high vacuum in the electron microscope for too long. Please edit.

Page 17, line 23: "section 4.3"

Page 18, lines 14-15: This statement regarding the association of sampling temperatures with actual ice nucleation temperatures should preface measurements in the discussion of methods.

Page 18, line 21: The reason that the authors believe that the current results are correct in regard to the lack of contribution of complex secondary particles and soot as IPRs (and thus INPs), and why the previous studies erred, should be summarized.

Page 18, lines 24-25: This statement regarding the composition of the secondary particle category also belongs in the methods material, which was painfully short in describing the different categories and their justification.

Page 18, line 28: Are the studies herein and those summarized in Knopf et al. (2018) for cloud activation temperatures in the same range?

Page 20, first paragraph discussion of "sea salt": This discussion was odd. I could take argument with the authors about the supposedly "controversial" nature of ice nucleation involving marine aerosols overall, but let me focus on lines 6-8. Unless the authors wish to reject clear evidence in the papers mentioned or in papers published since involving specific sampling of sea spray particles (none referenced here), the ice activity is clearly if not definitively associated with contained organics in many instances. It is not really a hypothesis that the salt itself is not the INP, so it is good that the authors will not "exclude" this fact.

---

## Author Comment (AC1) · 4 Sep 2018

Hammer et al. consider the properties of particles within ice crystals, aka ice particle residuals (IPRs), at the Jungfraujoch, a high altitude mountain site in Europe. For this work they use a published ice selective inlet aka 'Ice-CVI', and compare this to the total aerosol via a simultaneous heated inlet sample to provide all, aka 'total', particles. In total, they consider ~4k ICRs over 7 days and in the ~-10 to -20 deg C range (i.e., mixed phase clouds). These particles were deposited on substrates for off-line analysis with electron microscopy (EM). There are several papers over the last decade, many by the groups represented here describing such measurements at the Jungfraujoch so this is not altogether novel. It does represent important data in an area of atmospheric science that is currently in need of more information. My major concern, however, is that the way ice residuals are described seems very simplistic and not of use to the broader atmospheric science community. For example, what is a C-rich particle? How would someone using AMS or a modeler compare this to their understanding of the atmosphere? Since studies of this type have been published, again, many by this group, I believe they here have a responsibility to make the results more comparable, and therefore more useful, to the broader community. I therefore suggest the authors consider the following suggestions and, pending another review, that this paper could be published in ACP.

1. The literature seems to predominantly use the term 'ice residual' (IR) as opposed to IPR. Is there a reason the authors have suggested a new term? Is it different than IR? If so a comprehensive description – and difference from IR - needs to be made. As it reads it seems they are the same so, if there is no difference, could you please remain consistent with the literature term IR so as not to confuse the reader.
   - We now use the term ice residual (IR) in the paper.

2. Page 3 Line 8: The authors seem to suggest water vapour homogeneously nucleates ice at -38 deg C and 140% RH; but that is not correct. One must first have droplets. Please remove 'vapour or' from this sentence.
   - Removed as suggested

3.  Page 5 Line 5 & 6: From the intro, the authors 'assume' the IPRs are INPs, not 'consider'. This is a very important distinction. The authors discuss artifacts; thus they themselves show they cannot make the direct association implied by 'consider'.

    - We changed the sentence according to reviewer II: *"The campaign lasted for five weeks with the aim to investigate IRs from mixed phase clouds which may reflect the initial INPs active in the cloud."*

4.  Table 1: Most aerosol composition measurements show the presence of sulfate and SOA particles as a dominant, if not the dominant, aerosol, at least by number. Is the 'complex secondary particles' this class? This seems to be implied later in the Discussion but is never clearly stated. If so, even if it is an assumption, this needs to be stated for clarity and comparability to the literature. (note: this seems to be suggested on Page 10 but warrants more than 1 line of text).

    - We agree. A new column is added to table 1 to make it easier to understand our classification and particle groups.

*Table 1: Classification criteria and possible sources/ explanation for particle groups for both, total aerosol and ice particle residuals.*

| Group | Major elements | Morphology/ beam stability | Source/ particle explanation |
|---|---|---|---|
| Soot | C | Chain-like or more compact agglomerates of primary particles | *Combustion, black carbon* |
| C-rich | C | No soot morphology | *Organic aerosol, biomass burning**, biological** * |
| Complex secondary particles | No X-ray spectra or S-peak | Most particles evaporating, some relatively stable | *Sulphur rich secondary organic aerosol, might also contain a substantial fraction of nitrates and other organics* |
| Aged – sea salt | Na, S (sometimes small amount of Cl and Mg) | Relatively stable | *Marine aerosol, sea spray, might contain organics* |
| Mixed –sea salt | Na, S (sometimes small amount of Cl and Mg) + mineral composition | | *Marine aerosol mixed with mineral particles. Might contain organics.* |
| Ca-rich | Ca, C, O | | *Mineral particles, calcium carbonates e.g. calcite* |
| Ca-sulphate | Ca, S,O | | *Mineral particles, e.g. gypsum and anhydrite* |
| Silica | Si, O | | *Mineral particles, e.g. quartz* |
| Alumosilicate | Al, Si, O | | *Mineral particles, e.g. kaolinite* |
| Fe- alumosilicate | Al, Si, Fe, O | | *Mineral particles, e.g. almandine* |
| Other-alumosilicates | Variable amounts of Na, K, Ca, Si, Al, O, Ti and Fe | | *Mineral particles, e.g. feldspars, illite and smectite (montmorillonite)* |
| Metal/ metal oxides | Fe, O or Ti, O or Fe, Cr, Mn | Fly ash was detected as spherical particles | *Mineral particles like hematite, magnetite and rutile, or steel particles (alloys)* |
| Pb-rich | Pb, or Pb, Cl | Single particle or inclusions within particle | *Helicopters and small aircrafts, previously reported at Jungfraujoch* |
| Other | Particles which do not meet the classification criteria above | | |
| Alumina[*] | Al, O | | *Artefact, Ice-CVI* |
| Ni-rich[*] | Ni | | *Artefact, Ice-CVI* |
| Cu-rich[*] | Cu | | *Artefact, particle substrate* |
| Pure salt[*] | Na, Cl | | *Artefact, hypothesised from secondary ice processes e.g. crystal break-up, marine origin** * |

*Most likely contamination.  **Uncertain origin because the chemical characterisation and/or morphology was not the typical for this particle group.*

5. Similarly, biomass burning particles are often noted as being of abundance just below sulfates and SOA. Is this the C-rich class here? If so please state this. I stress here: while the authors are reporting what they observe with their EM technique, they also need to make it clear their classes relate to common particles types if they wish to publish in a journal such as ACP. Please make these comparisons.
   - See new table 1 (same as comment 4)

6. Similarly, what are Ca-rich particles? The discussion seems to suggest they are mineral dust? How are these associated with any common aerosol type?
   - See new table 1 (same as comment 4)

7. The authors discuss observations of biological IRs in the introduction. These don't appear on Table 1 or Figures 4 or 5. Are they not observed or is there an instrumental reason they can't be detected? Are they a subset of the C-rich category? This needs to be stated clearly in the paper, especially in Figure 5, as it goes directly to comparison to the literature on ice residual composition and in the location where Table 1 is described.
   - Primary biological particles are normally classified based on chemistry (C, O and tracer elements i.e., N, P, K) and morphology. We do not see the normal identification specifics in the C-rich group, but still, we cannot exclude that particles in this group might be of biological origin. We therefore added biological particles as a possible explanation of the C-rich group in the new table 1.

8. Page 7, Line 6. Please remove 'Unfortunately, only'. Data are what was collected.
   - Removed as suggested.

9. Sample S-2b looks rather like a local combustion event. This seems to be implied later in the discussion. If it is please state at the location of the Figure for clarity to the reader.
   - We state now that this is a combustion event in the figure caption.
     We changed figure caption to: *"Relative number abundance of the different particle groups within total aerosol samples. Sample S-2b shows a combustion event with air mass history form the Po Valley, and sample S-5b is influenced by an analytical artefact from particle loss of volatile particles."*

10. Given this, could you please present figure S2 at this location – move it into the main paper from the supplement - for the samples to give the origin some context? It is mentioned in the text in the discussion but really needs to be given at this location.
    - We moved Figure S2 to the main paper as suggested.

11. Sample S-5b looks rather like a mixed mineral and perhaps combustion event. Is this so? It again does not look like a clean troposphere. Is this so? If it is please state at the location of the Figure for clarity to the reader. Please also see last comment re: back-trajectories.
    - This sample cannot be explained by back-trajectories, but an analytical artefact from particle loss. We changed the figure caption to (same as comment 9): *Relative number abundance of the different particle groups within total aerosol samples. Sample S-2b shows a combustion event with air mass history form the Po Valley, and sample S-5b is influenced by an analytical artefact from particle loss of volatile particles.*

12. I am confused with Section 4.1, 'Methodological Problems'. I believe this all needs to do in the Experimental section (2), not held until after the data is presented. First, Figure S5 seems to indicate that most of the collected particles are artefact. Is this the case? If so please state the percentage in the Artefact section. Second the reader goes through the results but only after they are presented learns there are issues with the inlet and perhaps the EM data which appear to outnumber the real data by several factors. This is not a logical order. This all needs to be clearly stated and placed in Section 2, not held to Section 4.
    - We moved this section to methodology and added the following sentences to the start of the paragraph to make the contribution of artefacts clearer: "*The IR samples are heavily loaded with artefacts (40-78% of the particles – alumina, Ni-rich and pure salt) easily characterised and removed in further analysis. The Cu-rich particles are a part of the substrates and can in principle be found in both IR samples and total aerosol samples.*"

---

## Author Comment (AC2) · 4 Sep 2018

**General Comments**:**

This paper discusses the most recent evaluation of the composition of ice crystal residues as measured by an ice-selective counterflow virtual impactor inlet (CVI) at the Jungfraujoch experiment station. The broader community is surely interested in this work, in general and to understand if all measurement artifacts have been solved to the point that a consistent and informative data set can be collected regarding the source of ice nucleating particles and other information on microphysical processes in winter clouds at this site. The short answer based on this paper is that, while some issues have been resolved, this remains a work in progress. While a potentially useful paper, this one could have used stronger editing prior to submission. The overall organization is generally good, but the discussion bounces around such that the various facts are not subjected to a structured discussion. As for specific critical revisions needed, a vital one is to bring forward the fact that a major measurement artefact remains unresolved, that of Al in many particles. I saw no clear discussion of the potential source for this contamination. Otherwise, I appreciated the attempt to categorize electron microscopy microprobe data, although I stumbled on the categorization of "sea salt", by which was meant an assortment of possibilities. This pointed to a general need to be more descriptive about the categorizations and how artefacts were defined in comparison to specific sources. With revision, this paper will become acceptable for publication, although it remains another step in the direction of attempts to extract information on ice nucleation processes through inspection of ice particles residuals. Specific questions/comments for potentially addressing are listed below.

**Specific Comments:**

Abstract: A few details here should be clarified.

I have no idea what a multi MINI impactor is, but it certainly does not need listing in the abstract. Just the basic technique should be stated.

- A multi MINI impactor is a particle sampler with the possibility to sample on two different substrates at the same time with different particle cut off size, and to change sample without opening the impactor (for up to 12 samples) (Ebert et al., 2016). We agree that only the basic technique should be stated and will change the sentence to: *"Particles were sampled behind an ice selective counterflow impactor (Ice-CVI) for IRs and a heated total inlet for the total aerosol particles."*

It is not clear how a dilution setup allows for matching a total aerosol sample to the Ice-CVI sample. Can this be explained in plain language? Perhaps, "A dilution system was used to collect total particles at a similar rate to Ice-CVI particle collections", although I do not know how that is managed, and it seems that it failed about 50.

- The dilution factor is set before sampling and does not vary with the real time INP concentrations, and will hence not lead to matching the exact number of particles. The dilution is in use to be able to have a particle load on the substrate which allows for individual particle analysis. We changed the sentence to: *"A dilution setup was used to collect total particles with the same sampling duration as for IRs to prevent overloading of the substrates."*
- We changed the 2.1. "Sampling" section to contain a subsection of total aerosol sampling (ref. comment Page 3, paragraph 2).

State temperatures as "local" or "site". These are not necessarily the cloud activation temperatures.

- Changed as suggested: "*Temperature at the site*".
- We agree that the reported temperature is not necessarily the cloud activation temperature as we write in the discussion. Referring to a later comment, Page 18, lines 14-15: This statement regarding the association of sampling temperatures with actual ice nucleation temperatures should preface measurements in the discussion of methods. The statement is now moved to the method section. "*Temperatures was measured at the station, and can differ to the onset ice nucleation temperature of the particles depending on where in the mixed phase cloud nucleation occurred*".

"Approximately 3000 total aerosol particles from five days in clouds were also analysed." Is this referring to IPRs or to interstitial particles, or to all non-ice particles?

- 3000 total aerosol particles is referring to all particles, IRs and interstitial particles. We have now written: *"Approximately 3000 total aerosol particles (IRs and interstitial particles) from five days in clouds were also analysed."*

Introduction:

Page 3, line 8 – Spontaneous freezing of supersaturated vapour? In the Earth's atmosphere? I have not seen such a statement in the literature in some time. Remove unless you can support the feasibility of any process other than homogeneous freezing, not "homogeneous ice nucleation".

- Changed according to reviewer 1: removed "*vapour or*".

Page 3, lines 12 to 15 – Add "hypothesized" to modes. The last sentence is repetitive with regard to mixed-phase temperatures where heterogeneous nucleation is the source of ice initiation.

- Changed accordingly:
  *"Heterogeneous ice nucleation can occur in different hypothesized modes; (1) deposition nucleation, (2) immersion freezing, (3) contact freezing and (4) condensation freezing. A detailed description of the different modes are found elsewhere (Vali et al., 2015; Kanji et al., 2017).*
- Changed the last sentence to: *"Mixed phase cloud temperature range between -40°C and 0°C (Storelvmo, 2017), with immersion and contact freezing as the dominating ice formation modes (Lohmann and Diehl, 2006)."*

Page 3, paragraph 2 – The need for this thesis-type material is questionable. I suggest to revise and reduce or even omit most of this and get straight to the point, which seems to be that information on the relevance and importance of different ice nucleating particle types has come from laboratory measurement, and these emphasize the importance of mineral dust particles except at very modest cloud supercooling. What this paragraph does not seem to mention are specific studies where activated ice nucleating particles have been studied for composition, not simply tested as single collected types in the laboratory.

- We changed paragraph 3 and 4 in the introduction to:

[revised manuscript text omitted]

Page 4, line 4: Ogren et al. (1985) is not in the reference list. There is a substantial amount of literature since in which airborne CVIs have focused on ice clouds, and Cziczo, Froyd and colleagues have emphasized some other constraints on ice cloud sampling of IPRs (e.g., a focus on small ice, as done also in this study – line 6 statement, although no indication is given as to why fresh ice is needed). While an aside of sorts, the utility of sampling in mixed-phase clouds for ice nucleation studies using a non-ice-CVI is not reflected here, since the focus is on IPRs. The fact that one gets both IPRs and liquid cloud residuals when sampling in mixed-phase clouds is not necessarily a detriment, and this makes it suitable for ice nucleation measurements and subsequent collection of the activated INPs for compositional analyses (already mentioned in the preceding point). This is alluded to later in mentioning use of a FINCH for a similar purpose.

- See the new version of the paragraph in the comment above.
- We added Ogren et al (1985) to the reference list. Literature regarding characterisation and sample specifics with CVIs are additionally added to the references: *In-situ cloud measurements of ice particle residuals (IRs) can be done with an aircraft for pure ice clouds, like for cirrus clouds, with the use of a counter flow virtual impactor (CVI) (Ogren et al., 1985;Heintzenberg et al., 1996;Ström and Ohlsson, 1998;Twohy et al., 2003;Froyd et al., 2010;Cziczo and Froyd, 2014;Cziczo et al., 2017), and references therein)*
- We added an explanation to why fresh ice is important: *"A dedicated inlet system (Ice-CVI) was developed by Mertes et al. (2007) to sample freshly produced ice particles in mixed phase clouds and, after sublimating the ice, deliver the residuals (IRs) to connected sampling or analysing instruments. As described in Mertes et al. (2007), a residual particle can be interpreted as their original INP only when sampling small ice crystals. There are three reasons for this size restriction leading to sampling of rather young ice particles. First scientific reason is that only the small ice particles grows by water vapour diffusion. Larger ice particles could further grow by riming. Moreover, larger and older ice particles experience impaction scavenging by interstitial particles. Both processes add more aerosol particles to the ice crystal and thus the original INP cannot be identified any more after ice sublimation in the Ice-CVI. Last is a technical reason. Larger ice particles would shatter and break-up at the inner surfaces of the Ice-CVI sampling system."*

- We agree with the reviewer that collection of IRs and liquid cloud residuals are important in other cases. Due to the fact that the number density of supercooled droplets is much higher than of ice particles in a mixed-phase cloud, it is not possible to measure IR properties as long as the sampling is not ice selective.

Page 4, lines 8-10: But then this introduction is followed with these lines, which I could not understand - "Knowledge on particle groups acting as ice nuclei in mixed phase clouds is contradictory. IPRs are the residuals ice crystals formed on real INPs after they have been activated in the environment and the measured ice nucleation efficiency of these IPRs is then considered to be the same as for INPs." I expected the first sentence to be immediately supported. Is this a new paragraph? It is not a good one for sure. Rewrite it

to be concise, and get to that point. Is the contradiction mentioned referring only to studies done at Jungfraujoch, or what other studies? Will this study seek to resolve contradictions? What is a "real" INP? I suggest to remove this terminology. I think I understand the last part to mean that the composition of IPRs is considered to be those of INPs that were active at the local temperature of observation.

- We agree with the reviewer. The confusion in this paragraph starts with the wrong use of the word *"contradiction".* In this study, we think we are able to reach a good match between IR and INP because of our possibility to identify artefacts from the comparison of IR and total particles. As we already write this in the method, this part of the paragraph is now removed.

Experimental:

Page 5, lines 5-6: Please explain or omit the statement ". . .original true INPs." You will simply assume that IPRs represent INPs active at the cloud temperature of observation, correct? Are you trying to infer that other methods will not detect INPs? I think you are trying to say that the residuals reflect INPs that were activated in the cloud. But are you saying that every ice crystal contains an INP? I do not think that can be supported, if for example secondary ice formation processes were active.

- We corrected the sentence to: *"The campaign lasted for five weeks with the aim to investigate IRs from mixed phase clouds which may reflect the initial INPs active in the cloud."*
- Our intention is to sample primary small ice particles, which are freshly produced, but it is also possible to sample small fragments of the same size from secondary ice formation. These fragments could contain no aerosol particle, the INP (by chance), or scavenged particles. We cannot with our technique detect ice particles without a residual, and will therefore only know the chemical composition of the two latter. Sampling of secondary ice can explain part of the artefact particles.

Page 5, line 7: typo, "were" not "where"
- Corrected accordingly

Page 5, lines 9-10: Can you explain the need for dilution of the total aerosol sample a little better? i.e., there would be too many particles if collected for the entire time period?

- We changed the "2.1. Sampling" section to contain a subsection of total aerosol sampling (new section 2.2.) and added a sentence for explanation: *Without this dilution, due to the much higher concentration of total particles, these samples would be overloaded and not suited for single particle analysis."*

*"2.1. Sampling*

*In January/February 2017 an extensive field campaign was conducted by INUIT (Ice Nucleation Research Unit funded by the German Research Foundation DFG) at the*

*high altitude research station Jungfraujoch in Switzerland (3580m asl). The campaign lasted for five weeks with the aim to investigate IRs from mixed phase clouds which are considered as the original true INPs. During mixed phase cloud events, IRs where separated from other cloud constituents like interstitial aerosol particles, supercooled droplets and large ice aggregates by use of the Ice-CVI (Mertes et al., 2007). Total aerosol particles (interstitial particles and IRs) were sampled in parallel. Particles where sampled by the use of multi MINI cascade impactors with the same design as described in Ebert et al. (2016) and Schütze et al. (2017), but with the use of only one stage with a lower 50% cut-off diameter of approximately 0.1 µm (aerodynamic). The multi MINI cascade impactor is equipped with purge flow and 5 min flushing of the system was always performed prior to sampling to avoid carryover of particles from previous samples. The particles were collected on boron substrates to allow detection of light elements including carbon (Choël et al., 2005;Ebert et al., 2016)."*

*"2.2. Total aerosol sampling*

*Total aerosol particles were sampled in parallel to IRs behind a heated inlet (Weingartner et al., 1999) to study IR enrichment and depletion, identify contaminants and characterise the air-masses present. Total aerosol samples were collected with a dilution setup (Fig. 1) to match the longer sampling time (up to 5 hours) of the Ice-CVI. The dilution unit is build up by two valves to control the air stream in and out of the system, making it possible to send air through two filters to dilute the incoming aerosol flow. Without this dilution, due to the much higher concentration of total particles, these samples would be overloaded and not suited for single particle analysis. "*

Page 6, lines 18-19: Why are pure salt, alumina, Cu-rich and Ni-rich particles considered as contamination? It would be nice to consolidate this information in one place. In the end, no source is identified or even suggested for the alumina particles assumed as contamination, and I find the fresh salt explanation to be questionable. I gather later that the Al is assumed to come from ice crystals striking the walls of the CVI, despite coating them with Ni, but it is almost incomprehensible how this contamination exceeds that found in any previous study (page 15).

- We added the possible source of the contamination particles in the new table 1 as requested by reviewer 1.
- The Ni-coating of the Ice-CVI was ineffective as we still detect alumina-particles and only a small fraction of Ni-containing particles.
- Higher relative amount of contamination particles than previously found can be explained by different INP concentration, meteorological conditions, as well as particle load on the substrate and the sampling time. As we believe, alumina particles are from the system, these factors will influence the relative abundance of the different particle groups, which makes it difficult to compare to previous results.

We added to page 6 line 17: *"Classification criteria and possible sources are given in table 1".*

- We changed the sentence (ref. comment. Page 15, line 15-17): *"The relative abundance of alumina particles in IR samples is higher in our campaign compared to two previous campaigns at Jungfraujoch using the same instrumentation but without the Ni coating of the Ice-CVI (Ebert et al., 2011; Worringen et al., 2015). This might be explained by the fact that we only focused on sub-micrometer particles and/ or the difference in meteorology, sample time and particle load all influencing the relative composition of contamination particles."*

*Table 1: Classification criteria and possible sources/ explanation for particle groups for both, total aerosol and ice particle residuals.*

| Group | Major elements | Morphology/ beam stability | Source/ particle explanation |
|---|---|---|---|
| Soot | C | Chain-like or more compact agglomerates of primary particles | *Combustion, black carbon* |
| C-rich | C | No soot morphology | *Organic aerosol, biomass burning**, biological*** |
| Complex secondary particles | No X-ray spectra or S-peak | Most particles evaporating, some relatively stable | *Sulphur rich secondary organic aerosol, might also contain a substantial fraction of nitrates and other organics* |
| Aged – sea salt | Na, S (sometimes small amount of Cl and Mg) | Relatively stable | *Marine aerosol, sea spray, might contain organics* |
| Mixed –sea salt | Na, S (sometimes small amount of Cl and Mg) + mineral composition | | *Marine aerosol mixed with mineral particles. Might contain organics.* |
| Ca-rich | Ca, C, O | | *Mineral particles, calcium carbonates e.g. calcite* |
| Ca-sulphate | Ca, S,O | | *Mineral particles, e.g. gypsum and anhydrite* |
| Silica | Si, O | | *Mineral particles, e.g. quartz* |
| Alumosilicate | Al, Si, O | | *Mineral particles, e.g. kaolinite* |
| Fe- alumosilicate | Al, Si, Fe, O | | *Mineral particles, e.g. almandine* |
| Other-alumosilicates | Variable amounts of Na, K, Ca, Si, Al, O, Ti and Fe | | *Mineral particles, e.g. feldspars, illite and smectite (montmorillonite)* |
| Metal/ metal oxides | Fe, O or Ti, O or Fe, Cr, Mn | Fly ash was detected as spherical particles | *Mineral particles like hematite, magnetite and rutile, or steel particles (alloys)* |
| Pb-rich | Pb, or Pb, Cl | Single particle or inclusions within particle | *Helicopters and small aircrafts, previously reported at Jungfraujoch* |
| Other | Particles which do not meet the classification criteria above | | |
| Alumina[*] | Al, O | | *Artefact, Ice-CVI* |
| Ni-rich[*] | Ni | | *Artefact, Ice-CVI* |
| Cu-rich[*] | Cu | | *Artefact, particle substrate* |
| Pure salt[*] | Na, Cl | | *Artefact, hypothesised from secondary ice processes e.g. crystal break-up, marine origin*** |

*Most likely contamination. **Uncertain origin because the chemical characterisation and/or morphology was not typical for this particle group.

Page 7, line 5: This seems to require a statement that the cloud sampling temperatures were considered as appropriate as the ice crystal formation temperature. Could satellite data say anything about coldest cloud top temperatures at these times? Or do you also assume that the limited ice crystal size range sampled restricts this condition?

- We don't know the ice nucleation temperature, therefore, we can only give the site temperatures.
- We changed the sentence to: *"During seven days, ten Ice-CVI samples were taken in clouds at the site temperatures between -10 and -18°C. Sampling day, time and site temperatures are presented in Fig.2, and as table in the electronic supplement (table S1)".*
- We moved this sentence about site temperature to the method section (Ref. comment Page 18, lines 14-15) *"Temperatures was measured at the station, and can differ to the onset ice nucleation temperature of the particles depending on where in the mixed phase cloud nucleation occurred".*

Results:

Page 12: A general comment - it might be nice to show both a representative particle image and elemental spectra for each of the different particle composition categories. This could go in the supplement in addition to the single example given.

- We added particle examples to the supplement (see the supplement section below the comments).

Page 12: General comment 2 – It is only if one goes immediately to look at Fig. S5 at this point that one realizes that the vast majority of particles were categorized as artifacts. Surely this needs to be mentioned upfront. Greater that 50.

- We moved the artefact discussion to methods and added the relative amount of alumina particles accordingly to reviewer 1. We added the following sentences at the beginning of the paragraph: *"The IR samples are heavily loaded with artefacts (40-78% of the particles – alumina, Ni-rich and pure salt) easily characterised and removed in further analysis. The Cu-rich particles are a part of the substrates and can in principle be found in both IR samples and total aerosol samples".*

Page 12, line 7: I wonder if in the basic analysis performed if a mineral particle could be distinguished as being from desert or from other soils? I assume this would remain unresolved, since the soil particle could have multiple potential actual ice nucleation sources, including trace organics.

- Indeed an interesting question, but SEM-EDX is not suited for this analysis.

Page 12, line 10: When the authors say "sea salt", what is meant? Is it only NaCl, or does this refer to aerosols of sea spray origin, with a more complex mixing state? There are only two categories, aged and (aged-) mixed, and by mixed are also included mixtures with other aerosols such as minerals. This makes attribution specifically to "sea salt" nebulous, and yet statements are subsequently made in the results about the ice activity of "sea salt". This is problematic.

- We agree with the reviewer and changed the name of these particles to *sea salt-containing particles* – as commented later (Page 15, line 1). The particle groups are put together for statistical reasoning, with the assumption that they originate or at least both contain some material from the sea. We cannot say with our method what part of the particles that make them ice active.

Page 12, line 14: I assume that aluminum oxides are omitted from the metal oxide category because of the alumina contamination that is not really discussed?

- This is right. As we added a column to table 1 explaining the origin of the different particle groups, this point should be clear now.

Discussion:

Page 15, line 1: Sea salt is similarly ice active as aluminosilicates? Is it the sea salt, the organic content of marine aerosols, or the particles they are mixed with? Hence my earlier question. Perhaps these should be stated to be sea salt-containing particles, and a statement is needed about how this does not identify the "salt" as the ice nucleating component.

- We cannot identify with our method the main component for ice nucleation, and have to leave this as an open question. The pure salt is not present in the total aerosol samples and are hence regarded as contamination.
- We changed the name of sea salt (aged and mixed sea salt) accordingly to *"sea salt-containing particles"*.

Page 15, lines 15-17: A focus on fine particles is mentioned as an explanation for the occurrence of more alumina in this study, apparently from crystals etching this from the CVI walls (nowhere stated clearly). This is the first mention of any different focus in this study. What is meant be a focus on fine particles? Why would there be so much less Ni and so much more Al? Was the coating quickly destroyed? Ineffective? Also on line 15, "but" is misspelled.

- Most IR particles (>90 %) in our study are smaller than 1 µm (equivalent projected area diameter). Worringen et al. 2015 and Ebert et al. 2011 sampled with two stages, and one of them had a larger particle cut off (1 µm). The size distribution in Worringen et al. 2015 showed that one stage sampling is sufficient, and our size distribution is comparable to their size distribution in terms of IRs. One stage of the sampler had a 50 % cut off at 1 µm in the previous studies, which might exclude some of the alumina particles leading to a lower relative abundance of this group.
- Also the meteorology is different to previous campaigns.

- We corrected *"but"* accordingly.
- We rewrote the explanation to: *"The relative abundance of alumina particles in IR samples is higher in our campaign compared to two previous campaigns at Jungfraujoch using the same instrumentation but without the Ni coating of the Ice-CVI (Ebert et al., 2011; Worringen et al., 2015). This might be explained by the fact that we only focused on the sub-micrometer particles and/ or the difference in meteorology, sample time and particle load all influencing the relative composition of contamination particles."*

Page 15, lines 24-26: If pure or "fresh" salt is an artefactual reflection of secondary ice formation contributions, how is this reliably distinguished from sea spray aerosols? Would aging of sea salt always occur for marine particles reaching the site? Relatively unaged marine aerosols are found at other remote locations.
- We agree, we cannot exclude that this is of marine source. We have added marine as possible source in the new table 1.

Page 16, line 12: By "concentration of the total inlet" do you mean the accumulated particle number concentrations sampled from the total particle inlet (after dilution)?
- There were CPCs operating behind the total inlet, interstitial inlet and Ice-CVI during the campaign.
- We changed the sentence to*: "Three of the total aerosol samples (S-3b, S-4b and S-6b) are sampled under conditions where the concentration (measured with condensation particle counters) of the total inlet was lower than the interstitial inlet."*

Page 17, lines 8-10: This might well be the third mention of the sample that was exposed to high vacuum in the electron microscope for too long. Please edit.
- Changed to: *"The high soot and C-rich particle abundance of the first sample may be explained by footprint plots showing that the air-mass had a longer surface residence time over Po Valley (Italy) which is an urban/industrial area with abundant sources of carbonaceous particles. The potential artefact in the second sample does not influence the enrichment factor all other particle groups."*

Page 17, line 23: "section 4.3"
- Changed as suggested

Page 18, lines 14-15: This statement regarding the association of sampling temperatures with actual ice nucleation temperatures should preface measurements in the discussion of methods.
- Moved to method section as suggested.

Page 18, line 21: The reason that the authors believe that the current results are correct in regard to the lack of contribution of complex secondary particles and soot as IPRs (and thus INPs), and why the previous studies erred, should be summarized.

- Changed to: *"Complex secondary aerosol particles and soot were not found in the IR fraction, in contrast to previous work at Jungfraujoch (Cozic et al., 2008a;Ebert et al., 2011;Worringen et al., 2015;Schmidt et al., 2017), even though these groups dominate the total aerosol fraction. Thus, their ice nucleating ability under the conditions of our campaign, can be assumed to be very low. One explanation for this difference might be the higher site temperatures during our campaign."*

Page 18, lines 24-25: This statement regarding the composition of the secondary particle category also belongs in the methods material, which was painfully short in describing the different categories and their justification.

- We fully agree, a more descriptive table can now be found in the new table 1.
- We added a reference to table 1 in the sentence: "*It must be emphasised here that this particle group most likely also consists of a substantial fraction of organics and nitrates (Vester et al., 2007), see table 1."*

Page 18, line 28: Are the studies herein and those summarized in Knopf et al. (2018) for cloud activation temperatures in the same range?

- Changed to: "*C-rich particles are reported in previous studies of mixed phase clouds at Jungfraujoch (S.Mertes et al. 2007; Cozic et al. 2008; Kamphus et al. 2010; Ebert et al 2011; Worringen et al. 2015; Schmidt et al. 2017). Our results are also in agreement with findings of many cirrus cloud field studies (see recent review by Knopf et al. (2018) and references therein) which show that organic aerosol is found in the IR fraction, but is depleted relative to total aerosol.*"

Page 20, first paragraph discussion of "sea salt": This discussion was odd. I could take argument with the authors about the supposedly "controversial" nature of ice nucleation involving marine aerosols overall, but let me focus on lines 6-8. Unless the authors wish to reject clear evidence in the papers mentioned or in papers published since involving specific sampling of sea spray particles (none referenced here), the ice activity is clearly if not definitively associated with contained organics in many instances. It is not really a hypothesis that the salt itself is not the INP, so it is good that the authors will not "exclude" this fact.

- We agree with the reviewer. Our intention was not to reject the evidence published in papers regarding ice nucleation of sea salt/ sea spray particles. Our intention was only to express that we cannot say with our method where the nucleation occurred.
- We now write: *"Sea salt containing particles may act as an INP due to the presence of organics (Wilson et a. 2015; DeMott et al. 2016; Iwata and Matsuki, 2018). However, we cannot define with our measurement technique where the ice nucleation occur in a particle, i.e. pores or thin coating."*

**Supplement:**

**Examples of each particle group**

Secondary electron image and corresponding energy dispersive X-ray spectrum for a typical particle of each group are presented under. B indicates the X-ray peak from the boron substrate.

[Figure]

[Figure]

[Figure]

[Figure]

[Figure]

**References:**

[revised manuscript text omitted]